# Latent Space Robust Optimization of Neural Processes with Aligned Stratified Order-Statistic Loss Reduction

Qi Tao [* 1]   Jiarong Wen [* 1]   Jing Yang [2]   Guanlin Wu [3]   Zhang Kaiyu [1]   Yiqin Lv [1]   Wumei Du [1]   Xingxing Liang [4]
Qi Wang [1 5]

## Abstract

Importance-Weighted Neural Processes (IWNPs) provide a principled framework for probabilistic meta-learning by using multi-particle latent representations to approximate the marginal log-likelihood of task data tightly. However, this work reveals that the standard optimization of IWNPs suffers from the Matthew effect in the latent space, where high-likelihood particles dominate gradient signals. The neglect of lower-likelihood regions leads to poor tail-risk generation and unstable fast adaptation. While robust objectives such as $\text{CVaR}_{\alpha}$ can mitigate these risks, they often entail a trade-off that degrades average-case performance. This work proposes Order-Statistics Aligned Neural Processes (OS-NPs) to achieve latent space robust optimization without sacrificing average result. Specifically, we stratify multiple inference particles into disjoint difficulty bins based on order statistics and derive the regularized worst-case optimization framework for OS-NPs. Our method aligns the reduction of stratified order-statistic losses in IWNPs and provides a computationally efficient pipeline to implement. Extensive experiments demonstrate that the OS-NP constitutes stable, reliable probabilistic meta-learning that significantly enhances tail-risk robustness while maintaining or even improving average performance.

*Equal contribution [1]College of Sciences, National University of Defense Technology, Changsha, China [2]Naval Submarine Academy, Qingdao, China [3]Academy of Military Science, Peking, China [4]College of Systems Engineering, National University of Defense Technology, Changsha, China [5]Department of Automation, Tsinghua University, Peking, China. Correspondence to: Qi Wang <hhq123go@gmail.com>.

*Proceedings of the 43rd International Conference on Machine Learning*, Seoul, South Korea. PMLR 306, 2026. Copyright 2026 by the author(s).

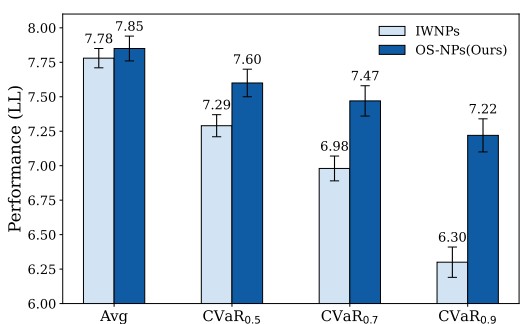

*Figure 1.* Average and $\text{CVaR}_{\alpha}$ Log-likelihoods of IWNPs and OS-NPs (Ours) in CIFAR-10 Image Completion (Foong et al., 2020). Here, we use 32 inference particles at the meta-test phase.

## 1. Introduction

Neural Processes (NPs) (Garnelo et al., 2018b) have emerged as a powerful meta-learning probabilistic framework. It enables efficient adaptation to new tasks with a few examples by leveraging knowledge from previously seen tasks (Hospedales et al., 2021). Through probabilistic mappings from context observations to target predictions, the NP captures the distribution over functions and quantifies the predictive uncertainty. This makes it particularly useful for learning stochastic processes while avoiding expensive computations such as predictive distribution in Gaussian processes (MacKay et al., 1998).

**Robustness to the inference particle matters in securing reliable fast adaptation at test-time.** Note that the vanilla variational inference optimization objective for NP leads to suboptimal inference and poor performance. This work focuses on the branch of Importance-Weighted Neural Processes (IWNPs) that require multiple particles from the latent space, e.g., convolutional neural processes (ConvNPs) (Foong et al., 2020). Even though importance-weighted optimization objective yields tighter likelihood estimates (Foong et al., 2020); the conditional generation results from these particles exhibit various qualities and sometimes suffer from low fidelity. As illustrated in Fig. 1, the conditional value-at-risk ($\text{CVaR}_{\alpha}$) of IWNPs' image completion performance across the population of tail-risk particles significantly degrades in terms of the log-likelihoods. The

vulnerability is critical in risk-sensitive applications, such as robotic systems using NPs as world models (Galashov et al., 2019), where low-quality generation tends to mislead policy optimization and guide dangerous actions. The realistic demand prompts us to study IWNPs through the lens of robust optimization (Levy et al., 2020) in the latent space.

**Let us alleviate the Matthew effect in importance-weighted objective through balanced optimization.** The IWNPs account for arbitrary sampling and provide an overall evaluation of generation; however, our theoretical analysis and empirical observations reveal that this objective tends to focus more on high-performing particles while ignoring others during optimization. Naively applying robust optimization, such as CVaR$_\alpha$ (Rockafellar et al., 2000), to the IWNP often leads to a trade-off: improving worst-case or tail-risk performance comes at the cost of reduced average fast adaptation performance. This raises a crucial research question: *Can we devise a feasible optimization strategy for IWNPs that enhances worst-case performance while hardly sacrificing average-case performance?*

This work provides a positive answer to the above question and presents a scheme that bins particles by stratified order statistics of adaptation results. It develops a regularized worst-case optimization framework, leading to Order-Statistics Aligned Neural Processes (OS-NPs). This manner can reconcile the gap between conventional likelihood-based performance and robustness-focused objectives.

**Contributions and Findings:** Our primary technical contribution is two-fold:

1. We identify a critical limitation of importance-weighted optimization in NPs, where gradients collapse to the best-performing particle, termed the Matthew Effect, and introduce fast adaptation robustness in the latent space.

2. To balance the worst-case adaptation and average performance, we craft a regularized worst-case optimization strategy and derive OS-NPs in a computationally efficient manner.

Extensive results demonstrate OS-NPs' superior performance across diverse tasks, illuminating both improved worst-case adaptation and maintained average predictive performance.

## 2. Preliminaries and Existing Dilemma

**Notations.** Let $p(\tau)$ represent a distribution over functions, with $\tau$ a function sample. The context dataset to learn functional priors is denoted by $\mathcal{D}_\tau^C = \{(x_i, y_i)\}_{i=1}^n$, while $\mathcal{D}_\tau^T = \{(x_i, y_i)\}_{i=1}^{n+m}$ defines the target dataset to predict.

### 2.1. Importance-Weighted Optimization Objective

The NP family is a probabilistic meta-learning method based on generative modeling.

**Generative Processes.** The vanilla NP follows an element-wise generation in Eq. (1). The predictive mean and variance are parameterized by $\theta$, yielding the joint distribution:

$$\rho(y_{1:n+m}) = \int p(z) \prod_{i=1}^{n+m} \mathcal{N}(y_i; \mu_\theta(x_i, z), \Sigma_\theta(x_i, z))dz, \quad (1)$$

where $z \in \mathbb{R}^d$ constitutes a distribution over functions.

For a set of diverse tasks denoted by $\mathcal{T}$, we can perform factorization and derive the optimization objective as:

$$p(\mathcal{D}_\mathcal{T}^T | \mathcal{D}_\mathcal{T}^C) = \prod_{\tau \in \mathcal{T}} \left[ \int p_\theta(\mathcal{D}_\tau^T | z) p_\phi(z | \mathcal{D}_\tau^C) dz \right] \quad (2a)$$

$$\max_{\theta, \phi} \mathcal{L}(\theta, \phi) := \sum_{\tau \in \mathcal{T}} \ln \left[ \int p_\theta(\mathcal{D}_\tau^T | z) p_\phi(z | \mathcal{D}_\tau^C) dz \right], \quad (2b)$$

where $p_\phi(z | \mathcal{D}_\tau^C)$ acts as a functional prior for each specific task $\tau \in \mathcal{T}$, and $p_\theta(\mathcal{D}_\tau^T | z)$ quantifies the generative distribution of target signals, e.g., the generated images' likelihoods conditioned on context pixels. For simplification, we focus on the objective function for an arbitrary task $\tau$, as in (Garnelo et al., 2018b). The implementation involves the task batch during both training and testing stages.

**Approximate ELBO in Vanilla NPs.** Note that there is no exact form of functional prior or posterior, Garnelo et al. (2018b) introduces a surrogate objective,

$$\mathcal{L}_{\text{NPs}}(\theta, \phi) = \mathbb{E}_{q_\phi(z | \mathcal{D}_\tau^T)} \left[ \ln p_\theta(\mathcal{D}_\tau^T | z) \right] - \\ D_{KL} \left[ q_\phi(z | \mathcal{D}_\tau^T) \| q_\phi(z | \mathcal{D}_\tau^C) \right], \quad (3)$$

where $\mathcal{L}_{\text{NPs}}(\theta, \phi)$ approximates the ELBO, and the approximate posterior $q_\phi(z | \mathcal{D}_\tau^T)$ and the approximate prior $q_\phi(z | \mathcal{D}_\tau^C)$ share the same neural architecture. These target data points are assumed conditionally independent, i.e., $p_\theta(\mathcal{D}_\tau^T | z) = \prod_{i=1}^{n+m} p_\theta(y_i | x_i, z)$.

**Maximum-Likelihood in IWNPs.** Instead of optimizing the broken ELBO in Eq. (3), Foong et al. (2020) propose the importance-weighted optimization objective as a direct Monte Carlo estimate of the marginal log-likelihood of evidence:

$$\mathcal{L}_{\text{IWNPs}}(\theta, \phi) = \ln \left[ \frac{1}{B} \sum_{b=1}^B \exp \left( \ln p_\theta(\mathcal{D}_\tau^T | z^{(b)}) \right) \right], \\ \text{with } z^{(b)} \sim p_\phi(z | \mathcal{D}_\tau^C) \text{ and } b = 1, \ldots, B, \quad (4)$$

where $B$ particles are sampled from the functional prior and the log-sum-exponential trick is used in evaluating the

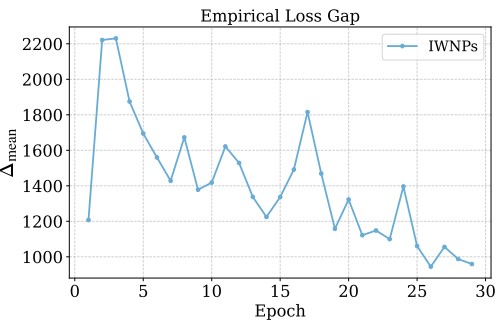

*Figure 2.* Empirical loss gap $\Delta_{\text{mean}} = \frac{1}{B}\sum_{b=1}^{B}\Delta^{(b)}$ of IWNPs during meta-training for the CIFAR-10 image completion.

multi-particle generation performance. From the empirical and theoretical analysis in (Foong et al., 2020; Wang et al., 2023), when one particle, i.e., $B = 1$, is used in Eq. (4), the optimization objective degenerates to the conditional neural process (Garnelo et al., 2018a).

## 2.2. Matthew Effect in Importance-Weighted Optimization

Note that $\mathcal{L}_{\text{IWNPs}}(\theta, \phi)$ softly aggregates the generation results from all latent samples. Compared with single-particle objectives (Garnelo et al., 2018a;b), the importance-weighted version (Foong et al., 2020) enjoys several benefits: (i) it explores different regions of the latent space simultaneously, representing distinct sub-populations or alternative contextual explanations; and (ii) multi-particle generation results provide a measure of robustness and stability.

For analysis simplicity, the following mainly discusses $\theta$ and quantifies the meta-test results in statistics. Given $z^{(b)} \sim p_\phi(z|\mathcal{D}_\tau^C)$, we express the per-particle loss, i.e., $\theta$-induced negative log-likelihoods as $\ell_{(b)} = -\ln p_\theta(\mathcal{D}_\tau^T|z^{(b)})$.

**Particles with high generation likelihoods dominate gradient flows.** Contrary to the potential benefits of importance-weighted optimization, our empirical observations in Fig. 1 imply that meta-test results diverge across the particle set. To this end, we conduct a diagnosis towards the phenomenon in Proposition 2.1 and attribute this to the severely imbalanced optimization in the latent space.

**Proposition 2.1** (Gradient Concentration in IWNPs). *Let $\ell_{b_{min}} = \min_b \ell_{(b)}$ be the minimum loss of the particle set, and we define the gap $\Delta^{(b)} = \ell_{(b)} - \ell_{b_{min}} \geq 0$. The gradient of $\mathcal{L}_{IWNPs}(\theta, \phi) = \ln\left[\frac{1}{B}\sum_{b=1}^{B}\exp(-\ell_{(b)})\right]$ regarding $\theta$ is a weighted average:*

$$\nabla_\theta \mathcal{L}_{IWNPs}(\theta, \phi) = -\nabla_\theta \ell_{b_{min}}$$
$$- \sum_{b \neq b_{min}} \frac{\exp\left(-\Delta^{(b)}\right)}{1 + \sum_{b' \neq b_{min}}\exp\left(-\Delta^{(b')}\right)}\left[\nabla_\theta \Delta^{(b)}\right]. \quad (5)$$

*As the gaps $\Delta^{(b)} \to \infty$ for all $b \neq b_{min}$, the weights*

$w_{b \neq b_{min}}$ *vanish at an exponential rate, causing the collapse to the single best particle:* $\nabla_\theta \mathcal{L}_{IWNPs} \approx -\nabla_\theta \ell_{b_{min}}$.

As noted, the gradient of the IWNP objective will be heavily biased toward the single best-performing particle $\ln p_\theta(\mathcal{D}_\tau^T|z)$ when its exponential term dominates the summation value. This work refers it to the *Matthew effect* (Merton, 1988) in IWNPs. As the performance gap between particles maintains a large value, e.g., Fig. 2, the IWNP degenerates into a max-operator, causing a dynamics focused on optimizing the current best particle.

## 2.3. Challenges in Latent Space Robust Optimization

The previous analysis reveals that IWNP tends to under-optimize particles with the worst generation results under certain scenarios. A straightforward solution to alleviate the Matthew effect is to perform distributionally robust optimization (Levy et al., 2020) in the latent space $p_\phi(z|\mathcal{D}^C)$. By shifting the objective from maximizing the average likelihood to minimizing the risk associated with the worst-performing particles, we can enforce a more equitable distribution of gradient information and keep the model expressive across the entire latent manifold.

**Definition 2.1** (CVaR$_\alpha$-induced Generative Robustness). *Let $\alpha \in (0,1)$ be a confidence level. Define the Value-at-Risk (VaR) at level $\alpha$ as the smallest threshold $\ell$ such that the loss exceeds $\ell$ with probability at most $1 - \alpha$:*

$$\text{VaR}_\alpha[\ell] := \inf\{\ell | \mathbf{Pr}_{z^{(b)} \sim p_\phi(z|\mathcal{D}_\tau^C)}\left[\ell_{(b)} \leq \ell\right] \geq \alpha\}. \quad (6)$$

*Conditional Value-at-Risk at level $\alpha$, i.e., CVaR$_\alpha[\ell]$ is the expected loss conditioned on being in the worst $(1 - \alpha)$ tail of the distribution (Rockafellar et al., 2000):*

$$\ln\left[\frac{1}{1-\alpha}\mathbb{E}_{p_\phi(z^{(b)}|\mathcal{D}_\tau^C)}\left[\exp(-\ell_{(b)})|\ell_{(b)} \geq \text{VaR}_\alpha[\ell]\right]\right]. \quad (7)$$

**CVaR-NPs.** Definition 2.1 induces the concept of latent space robustness in IWNPs, which also corresponds to the optimization objective of CVaR-NPs. And its practical implementation is the sample average Monte Carlo estimate of CVaR$_\alpha[\ell]$:

$$\mathcal{L}_{\text{CVaR-NPs}} = \ln\left[\frac{1}{(1-\alpha)\hat{B}}\sum_{b=1}^{\hat{B}}\omega(z^{(b)})\exp(-\ell_{(b)})\right], \quad (8)$$

where $\omega(z^{(b)}) = \mathbb{1}[\ell_{(b)} \geq \bar{\text{VaR}}_\alpha[\ell]]$ denotes the indicator function with $\bar{\text{VaR}}_\alpha[\ell]$ the estimated $(1 - \alpha)$ loss quantile. And Eq. (8) picks up $(1 - \alpha)\hat{B}$ worst particles to optimize.

Though applying CVaR$_\alpha$ optimization to IWNPs in the latent space is straightforward and theoretically sound in improving the adaptation robustness, several accumulated several evidences in other fields (Greenberg et al., 2023; Wang

et al., 2024b; Lv et al., 2024) imply that such optimization mostly degrades the average performance. In other words, striking a balance between average and worst-case adaptation performance creates an optimization dilemma.

## 3. Method

Recognizing the dilemma of robust optimization in IWNPs, this section investigates feasible strategies to balance the performance of various particles. We then provide a theoretical analysis and implementation pipelines.

### 3.1. Stop Rolling Dice of Particles to Optimize

Technically, CVaR-NPs emphasize a fixed $\alpha$-quantile of the loss distribution while the IWNP focuses on the maximum-likelihood particle. Both result in negligible updates for other particles.

To enhance robustness against tail particles while maintaining average performance, it is important to go beyond the optimization of a single extreme statistic. As illustrated in Fig. 3, we propose aligning robust optimization with the principles of multi-task learning (Caruana, 1997), treating each quantile interval expectation of the losses as a distinct yet interconnected optimization objective.

**Stratified Order-Statistics as Multi-Tasks.** Optimization over latent variables in IWNPs can be approximated using a set of $B$ particles $\{z_{(b)}\}_{b=1}^{B}$ and losses $\{\ell_{(b)}\}_{b=1}^{B}$. The associated order statistics $\ell_{\lceil 1 \rceil} \leq \cdots \leq \ell_{\lceil B \rceil}$ characterize different quantiles of adaptation difficulty.

Next we partition the empirical loss distribution into $K$ quantile-based strata, namely histogram binning (Wand, 1997). We first determine empirical quantile thresholds $c_k = \ell_{(\lceil kB/K \rceil)}$ based on the sorted losses. Accordingly, this yields $K$ disjoint subsets $\{S_k\}_{k=1}^{K}$, where

$$S_k = \{b | c_{k-1} < \ell_{\lceil b \rceil} \leq c_k\}, \quad \text{with } |S_k| \approx B/K. \quad (9)$$

Each stratum corresponds to a specific difficulty range in the latent space, from easy to extremely hard.

With inference particles grouped into quantile-based strata, optimization can be interpreted through a multi-objective lens. Each stratum represents a distinct task corresponding to a specific region of the loss distribution.

**Recast Importance-Weighted Optimization Objective to Multi-Objective Optimization.** Given the loss induced probability density function $p(\ell)$ and quantile function $q_\alpha$, we can express the quantile-interval expectation for any $0 \leq \alpha_1 < \alpha_2 \leq 1$ as:

$$\mathcal{L}_k(\theta) = \frac{1}{\alpha_2 - \alpha_1} \int_{q_{\alpha_1}}^{q_{\alpha_2}} \ell p(\ell) d\ell \approx \frac{1}{|S_k|} \sum_{b \in S_k} \ell_{\lceil b \rceil}, \quad (10a)$$

where Eq. (10a) defines the empirical stratum-wise loss.

Under this formulation, each quantile stratum focuses on optimization over specific regions of $p(\ell)$. Putting all strata together formulates a vector-valued objective in the form of multi-objective optimization problem (Deb et al., 2016):

$$\min_\theta \boldsymbol{L}(\theta) = [\mathcal{L}_1(\theta), \ldots, \mathcal{L}_K(\theta)]^\top, \quad (11)$$

where the solution concept corresponds to Pareto optimality that balances performance across quantile strata.

### 3.2. Kill Many Birds with One Stone

Before performing robust yet balanced optimization, we must not only calculate the likelihoods but also track the performance improvements of each stratum over iterations.

Now let $d^t$ denote the descent direction at $t$-th iteration, the decoder update of IWNPs relies on the merged gradient:

$$\theta^{t+1} = \theta^t - \eta d^t, \text{ with } d^t = \sum_{i=1}^{K} \alpha_i \nabla_\theta \mathcal{L}_i(\theta). \quad (12)$$

The selection of an appropriate $d^t$ is vital for improving the loss across all strata, particularly the most difficult one, thereby aiding in robust optimization.

**Robust Optimization of Stratum-wise Performance Improvement.** Naturally, we can perform the first-order Taylor expansion with respect to $\mathcal{L}_k(\theta^t)$ around $\theta^t$ and combine it with Eq. (12):

$$\mathcal{L}_k^{t+1} = \mathcal{L}_k^t - \eta(\nabla_\theta \mathcal{L}_k^t)^\top d^t + \mathcal{O}(\eta^2) \quad (13a)$$

$$\Leftrightarrow \mathcal{L}_k^t - \mathcal{L}_k^{t+1} = \eta(\nabla_\theta \mathcal{L}_k^t)^\top d^t + \mathcal{O}(\eta^2), \quad (13b)$$

where $\mathcal{L}_k^t - \mathcal{L}_k^{t+1}$ measures the progress of loss decrease for the $k$-th objective. Hence, maximizing the one-step loss decrease is roughly the same as maximizing $(\nabla_\theta \mathcal{L}_k^t)^\top d^t$.

Performing the worst-case optimization among all strata reduces to seeking the best merged descent direction:

$$\max_{d^t \in \mathbb{R}_m} \min_{k \in [K]} (\nabla_\theta \mathcal{L}_k^t)^\top d^t - \frac{1}{2}\|d^t\|^2, \quad (14)$$

where the quadratic penalty $\frac{1}{2}\|d^t\|^2$ helps rule out solutions with arbitrarily large magnitude. This formulation shares the same form as the multi-task gradient manipulation methods such as MGDA (Désidéri, 2012; Sener & Koltun, 2018).

As noted in (Liu et al., 2023a), Eq. (14) can be equivalently translated into seeking descent directions with a convex combination of stratum-wise gradients. This results in a quadratic programming problem:

$$\min_{\alpha \in \Delta^{K-1}} \frac{1}{2}\|G\alpha\|_2^2 \quad (15a)$$

$$\Rightarrow \min_{\xi \in \mathbb{R}^K} \frac{1}{2}\|G\alpha\|_2^2, \text{ with } \alpha = \text{softmax}(\xi), \quad (15b)$$

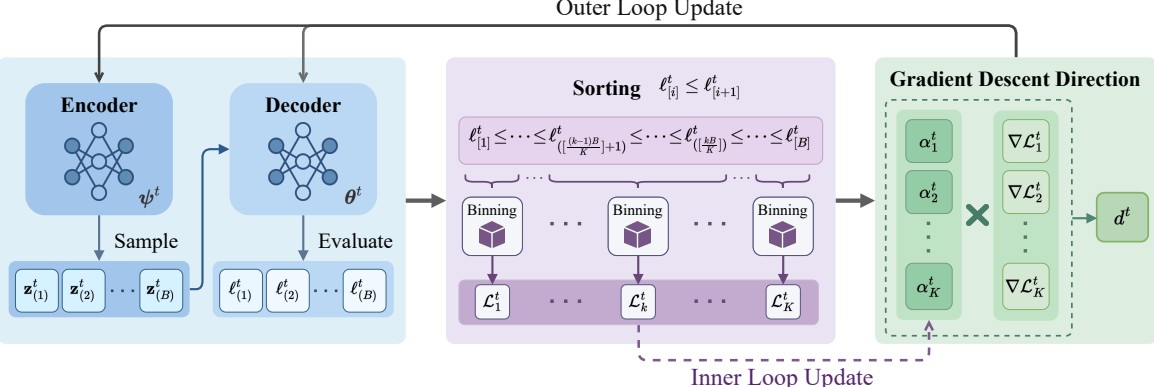

*Figure 3.* Optimization Pipeline of OS-NPs. Same as IWNPs, OS-NPs sample multiple particles and sort them according to the evaluated likelihoods of predictive distributions. The inner loop bins these results and solves the regularized worst-case optimization problem in an amortized manner. The outer loop updates the encoder-decoder parameters of NPs modules.

where $G = [\nabla_\theta \mathcal{L}_1, \ldots, \nabla_\theta \mathcal{L}_K]^\top$ denotes the stratum-wise gradients, and $\Delta^{K-1}$ is a simplex. Also, the practical objective as Eq. (15b) incorporates the unconstrained softmax logits $\xi$ to ensure that $\alpha$ remains within the simplex $\Delta^{K-1}$.

**Regularized Worst-Case Optimization in the Latent Space.** However, our empirical trials suggests that the challenge persists when solving Eq. (15b) to obtain the stratum weight for IWNPs' gradient update, as the auxiliary variable $\xi$ often retains large magnitudes. The highly peaked softmax weights prioritize a single stratum and suppress contributions from others. This violates the purpose of achieving balanced improvements across multiple strata.

To this end, this work introduces a regularized robust optimization objective as the *inner-loop* optimization problem in OS-NPs:

$$\min_{\xi \in \mathbb{R}^K} \mathcal{J}(\xi) \triangleq \frac{1}{2}||G\alpha||_2^2 + \frac{1}{2}\lambda||\xi||_2^2, \text{ with } \alpha = \text{softmax}(\xi).$$
$$(16)$$

Such a formulation maintains a focus on improving the performance of the worst stratum while penalizing gradient concentration via the $||\xi||_2^2$ term. As analyzed in Sec. 3.3, the regularized form induces strict strong convexity in the dual objective, securing the optimization remains well-posed. The following experimental results will further demonstrate that our design effectively accommodates the IWNP and that the optimization result is superior to the existing SoTA variant of the MGDA method.

**Gradient Aggregation with Amortized Optimization.** In solving Eq. (16), directly computing $\nabla_\xi \mathcal{J}(\xi) = \frac{\partial \alpha}{\partial \xi}^\top G^\top G \alpha + \lambda \xi$ raises additional computational overhead. The explicit construction of the Gram matrix $G^\top G$ requires $K$ backpropagation passes and $K^2$ pairwise inner products, leading to an $\mathcal{O}((K^2 + K)d)$ complexity per iteration.

Hence, we turn to an amortized approximation that leverages temporal continuity. By applying he Taylor expansion result in Eq. (13) and substituting the parameter displacement $\Delta\theta^{t+1} = -\eta G^t \alpha^t$, we derive an efficient estimator for the gradient product:

$$\left[(G^t)^\top G^t \alpha^t\right]_k \approx \frac{\mathcal{L}_k^t - \mathcal{L}_k^{t+1}}{\eta} \triangleq \Delta\mathcal{L}_k^{t+1}. \quad (17)$$

This surrogate replaces expensive matrix operations with observed loss decay rates. The relative optimal $\xi$ for $(t+1)$-th iteration are then updated via *one-step gradient descent*, and we can derive the merged gradient as $d_*^{t+1}$:

$$\xi^{t+1} = \xi^t - \beta \left[ \frac{\partial \alpha^t}{\partial \xi}^\top \Delta\mathcal{L}^{t+1} + \lambda\xi^t \right] \quad (18a)$$

$$\alpha^{t+1} = \text{softmax}(\xi^{t+1}) \quad (18b)$$

$$d_*^{t+1} = \sum_{k=1}^K \alpha_k^{t+1} \nabla_\theta \mathcal{L}_k^t, \quad (18c)$$

where $\beta$ denotes the learning rate for optimizing the sub-programming problem. Note that the OS-NP pipeline executes Eq. (18a) once per iteration as the solution to Eq. (16).

### 3.3. Practical Implementation and Theoretical Analysis

OS-NPs adopt a bi-level optimization pipeline, with the *inner loop* estimating the optimal merged gradient direction and the *outer loop* updating encoder-decoder parameters in the standard IWNPs manner (See details in Algorithm 1).

**Assumption 1** (*L*-Lipschitz in Gradient). Each stratum-wise loss $\mathcal{L}_k$ satisfies over iteration:

$$||\nabla_\theta \mathcal{L}_k - \nabla_{\theta'} \mathcal{L}_k|| \leq L||\theta - \theta'||, \ \forall \{\theta, \theta'\} \in \Theta. \quad (19)$$

**Assumption 2** (Bounded Softmax Logits). There exists $B_\xi > 0$ s.t. $\xi^t$ in Eq. (16) satisfies $||\xi^t|| \leq B_\xi$ over iteration.

**Regularization and Convergence Analysis.** As established in Theorem C.3, $\mathcal{J}(\xi)$ in Eq. (16) exhibits global strong convexity when $\lambda > 2F^2$ (Boyd & Vandenberghe, 2004). This property addresses the lazy update issue in vanilla MGDA, where a singular Hessian can cause optimization stagnation. With the provided positive curvature, the involvement of $\|\xi\|_2^2$ ensures a unique weight assignment $\alpha^*$ and a consistent descent direction even in degenerate gradient areas.

**Theorem 3.1** (Existence and Uniqueness). *Suppose the task gradients are bounded, i.e., $\max_k \|\nabla_\theta \mathcal{L}_k\|_2 \leq F$. For any $\lambda > 2F^2$, $\mathcal{J}(\xi)$ is strictly strongly convex over $\mathbb{R}^K$. Consequently, the optimal solution $\xi^*$ exists and is unique, yielding a well-defined weight assignment $\alpha^* = softmax(\xi^*)$.*

Despite the bi-level design's breakdown of the vanilla gradient descent process and the introduction of an optimality gap via amortized optimization in the inner loop, we demonstrate in Theorem 3.2 that our optimization pipeline converges to some extent.

**Theorem 3.2** (Convergence of Stratified Gradient Descent). *Under Assumptions 1/2 along with the step-size conditions $\sum_{t=1}^\infty \eta^t = \infty, \sum_{t=1}^\infty (\eta^t)^2 < \infty, \sum_{t=1}^\infty \beta^t < \infty$, Algorithm (1) ensures that $\liminf_{t\to\infty} \|d^t\|^2 = 0$ in Eq. (18).*

---

**Algorithm 1** Stratified Optimization for IWNPs.

**Input** : Learning rate $\beta$ and $\eta$; Iterations $N_{\text{iter}}$; Coefficient $\lambda$; Particle and strata number $B$ and $K$; Initialized $\xi_0$.
**Output** : Meta-trained model parameters $\theta$ and $\psi$.

1   Initialize the model parameters $\theta^0$ and $\psi^0$;
2   Initialize:
3     Sample $B$ particles and form $K$ groups of size $[\frac{B}{K}]$;
4     Compute initial stratum-wise loss $\mathcal{L}^0$ from initial particles;
5     $\alpha^0 \leftarrow softmax(\xi^0)$;
6     Gradient Update:
7      $\theta^1 \leftarrow \theta^0 - \eta \sum_{k=1}^K \alpha_k^0 \nabla_\theta \mathcal{L}_k^0$;
8      $\psi^1 \leftarrow \psi^0 - \eta \sum_{k=1}^K \alpha_k^0 \nabla_\psi \mathcal{L}_k^0$;
9   **for** $t = 1$ **to** $N_{iter}$ **do**
     // Compute Robust Yet Balanced Direction
10     Sample $B$ particles and form $K$ groups of size $[\frac{B}{K}]$;
11     Compute stratum-wise loss $\mathcal{L}^t$;
12     Compute $\Delta\mathcal{L}^t = (\mathcal{L}^{t-1} - \mathcal{L}^t)/\eta$;
13     Gradient Update:
14      $\xi^t \leftarrow \xi^{t-1} - \beta\left(\frac{\partial \alpha^{t-1}}{\partial \xi}^\top \Delta\mathcal{L}^t + \lambda\xi^{t-1}\right)$;
15      $\alpha^t \leftarrow softmax(\xi^t)$;
     // Optimize Encoder-Decoder Parameters
16     Gradient Update:
17      $\theta^{t+1} \leftarrow \theta^t - \eta \sum_{k=1}^K \alpha_k^t \nabla_\theta \mathcal{L}_k^t$;
18      $\psi^{t+1} \leftarrow \psi^t - \eta \sum_{k=1}^K \alpha_k^t \nabla_\psi \mathcal{L}_k^t$.
19   **end**

---

**Computational Complexity.** OS-NPs match standard IWNPs by utilizing gradient linearity and Softmax Jacobian properties. Combining stratum-wise losses into a weighted scalar enables a single backward pass, maintaining $\mathcal{O}(BM)$ complexity with $B$ particles and $M$ network pass cost. The auxiliary costs for sorting particles ($\mathcal{O}(B \log B)$) and updating logit parameters ($\mathcal{O}(K)$) are minimal compared to the network pass. Thus, total complexity is $\mathcal{O}(BM + B \log B + K)$, introducing only marginal overhead during meta-training while retaining the fast adaptation efficiency of standard IWNPs at test time. Additional runtime analysis is provided in Appendix D.

# 4. Experiment and Analysis

This section mainly answers the primary research question in Section 1. In detail, we examine the influence of the order-statistics optimization strategy and compare the resulting OS-NPs with other SoTA methods.

**Baselines and Evaluation.** The SoTA IWNP (Foong et al., 2020) works as the backbone for all methods. We focus on enhancing performance in the tail of the predictive distribution using robust optimization objectives, including Monte-Carlo CVaR$_\alpha$, Group Distributionally Robust Optimization (GDRO) (Sagawa* et al., 2020), and Temperature-regularized Distributionally Robust Optimization (TDRO) (Gladin et al., 2025). These objectives modify the ConvNPs' training to yield IWNPs, CVaR-NPs, GDRO-NPs, and TDRO-NPs as baselines.

Throughout all experiments, we use a fixed number of particles for optimization and meta-testing. Specifically, we set $B = 16$ for optimization and $B = 32$ for meta-testing. In terms of metrics, we evaluate the predictive log-likelihoods of multi-particles, detailed in Eq. (8), (Le et al., 2018; Foong et al., 2020), and test the generative robustness by varying the confidence level of CVaR$_\alpha$, detailed in Eq. (8). This quantifies the risk associated with particle-induced randomness.

## 4.1. Synthetic Regression

We train models on Gaussian process (GP) functions generated from (i) an RBF kernel, (ii) a Matérn-$\frac{5}{2}$ kernel, and (iii) a Periodic kernel.

Table 1 indicates that OS-NPs set a new SoTA by effectively balancing average log-likelihood with superior tail-risk robustness. Models like IWNPs and GDRO-NPs degrade sharply at higher risk levels, whereas OS-NPs remain stable. For the RBF kernel, OS-NPs improve CVaR$_{0.9}$ by 16% over GDRO-NPs and 47% over IWNPs. On the Matérn-5/2 kernel, the gain is even more significant. These gains stem from our regularized gradient-balancing mechanism, which outperforms CVaR-NPs and TDRO-NPs across all metrics.

The visualization results in Fig. 5 further confirm these findings and reveal additional benefits of OS-NPs, including

*Table 1.* Predictive log-likelihood results on 1D synthetic regression tasks. We report the average performance and CVaR at different confidence levels (standard deviations over 5 runs). The **Best Results** are marked in bold with the underline ones as the Runner-Ups.

| Method | Matérn-$\frac{5}{2}$ | | | | Periodic | | | | RBF | | | |
|---|---|---|---|---|---|---|---|---|---|---|---|---|
| | Avg | $\text{CVaR}_{0.5}$ | $\text{CVaR}_{0.7}$ | $\text{CVaR}_{0.9}$ | Avg | $\text{CVaR}_{0.5}$ | $\text{CVaR}_{0.7}$ | $\text{CVaR}_{0.9}$ | Avg | $\text{CVaR}_{0.5}$ | $\text{CVaR}_{0.7}$ | $\text{CVaR}_{0.9}$ |
| IWNPs | $0.18_{\pm0.01}$ | $0.10_{\pm0.01}$ | $0.08_{\pm0.01}$ | $0.04_{\pm0.01}$ | $\underline{-0.80}_{\pm0.02}$ | $\underline{-0.92}_{\pm0.03}$ | $\underline{-0.96}_{\pm0.03}$ | $-1.03_{\pm0.04}$ | $\mathbf{0.56}_{\pm0.01}$ | $0.44_{\pm0.02}$ | $0.40_{\pm0.02}$ | $0.34_{\pm0.03}$ |
| CVaR-NPs | $0.17_{\pm0.02}$ | $\underline{0.16}_{\pm0.02}$ | $\underline{0.16}_{\pm0.02}$ | $\underline{0.15}_{\pm0.02}$ | $-0.96_{\pm0.09}$ | $-0.98_{\pm0.01}$ | $-0.99_{\pm0.01}$ | $\underline{-1.00}_{\pm0.01}$ | $0.51_{\pm0.02}$ | $\underline{0.50}_{\pm0.02}$ | $\underline{0.50}_{\pm0.02}$ | $\underline{0.50}_{\pm0.02}$ |
| GDRO-NPs | $\mathbf{0.23}_{\pm0.03}$ | $0.12_{\pm0.03}$ | $0.08_{\pm0.04}$ | $0.03_{\pm0.05}$ | $-0.85_{\pm0.08}$ | $-0.99_{\pm0.01}$ | $-1.04_{\pm0.01}$ | $-1.13_{\pm0.01}$ | $\underline{0.55}_{\pm0.01}$ | $0.46_{\pm0.01}$ | $0.44_{\pm0.02}$ | $0.43_{\pm0.01}$ |
| TDRO-NPs | $0.15_{\pm0.03}$ | $0.13_{\pm0.03}$ | $0.12_{\pm0.03}$ | $0.11_{\pm0.03}$ | $-0.82_{\pm0.01}$ | $-1.02_{\pm0.04}$ | $-1.09_{\pm0.05}$ | $-1.22_{\pm0.07}$ | $0.49_{\pm0.02}$ | $0.48_{\pm0.02}$ | $0.48_{\pm0.02}$ | $0.47_{\pm0.02}$ |
| OS-NPs(Ours) | $\underline{0.19}_{\pm0.01}$ | $\mathbf{0.17}_{\pm0.01}$ | $\mathbf{0.17}_{\pm0.01}$ | $\mathbf{0.16}_{\pm0.01}$ | $\mathbf{-0.79}_{\pm0.02}$ | $\mathbf{-0.87}_{\pm0.03}$ | $\mathbf{-0.90}_{\pm0.03}$ | $\mathbf{-0.94}_{\pm0.04}$ | $0.52_{\pm0.02}$ | $\mathbf{0.51}_{\pm0.02}$ | $\mathbf{0.51}_{\pm0.02}$ | $\mathbf{0.50}_{\pm0.02}$ |

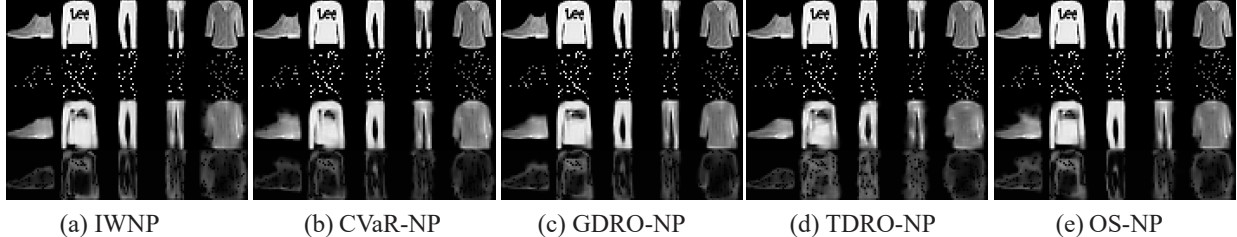

(a) IWNP     (b) CVaR-NP     (c) GDRO-NP     (d) TDRO-NP     (e) OS-NP

*Figure 4.* Visualization of FMNIST image completion using the worst-performing particle. From top to bottom are the ground truth, the context points, the reconstructed images conditioned on the worst-performing particle, and the corresponding predictive variance.

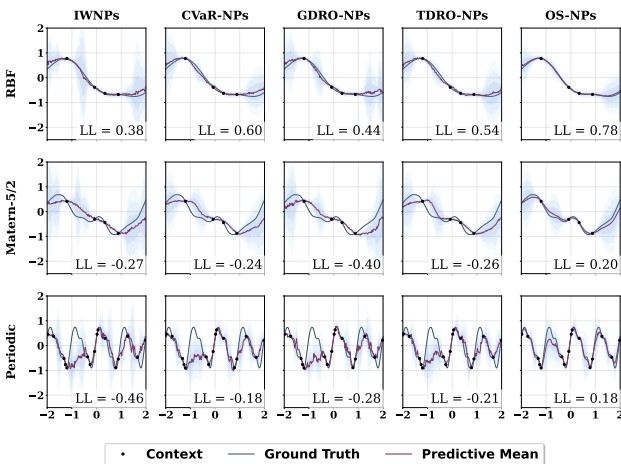

*Figure 5.* Predictive Distributions of Models under Worst-Performing Particles on Various GP Dataset. Illustrated are the mean and $\pm 3$ standard deviations.

more accurate predictive distributions and more reliable uncertainty quantification compared to baselines. The model excels at capturing both the smooth RBF kernel and the Matérn$-\frac{5}{2}$ kernel. Particularly, no other methods except OS-NPs can adequately cover the ground-truth curve within the uncertainty region in Periodic kernels.

### 4.2. Image Completion

We evaluate all methods on widely used datasets: FMNIST (Xiao et al., 2017), CIFAR-10 (Krizhevsky et al., 2009), and SVHN (Sermanet et al., 2012).

In Table 2, OS-NPs significantly outperform the IWNPs and TDRO-NPs baselines on the SVHN and CIFAR-10 datasets, particularly in tail-risk metrics. On SVHN, OS-NPs improve $\text{CVaR}_{0.9}$ by 59.3% over IWNPs and 5.3% over TDRO-NPs, which is the strongest baseline. While TDRO-NPs show robustness on SVHN, they perform poorly on FMNIST and lack consistency across data distributions. CVaR-NPs improve tail-risk performance but face mode collapse in latent space. OS-NPs, however, maintains reliability in the particle distribution while ensuring high-fidelity reconstruction.

As illustrated in Fig. 4, OS-NP maintains a sharp and semantically accurate reconstruction, whereas other baselines exhibit varying levels of blur. Additionally, our predictive uncertainty effectively identifies object boundaries. In contrast, vanilla IWNPs struggles with both reconstruction quality and uncertainty quantification. We further observe that OS-NPs also preserve competitive best-particle performance; additional visualizations and top-performing particle evaluations are provided in Appendix D.

### 4.3. Sim2Real

To examine out-of-distribution (OOD) generalization, we employed a Sim2Real protocol (Gordon et al., 2019). Models were trained solely on synthetic trajectories from the Lotka–Volterra simulator (Wilkinson, 2018; Papamakarios & Murray, 2016) and then evaluated on the Hudson's Bay lynx–hare dataset (Leigh, 1968) for interpolation based on sparse context points. As reported in Table 3, OS-NPs consistently achieve the highest log-likelihoods across all met-

*Table 2.* Predictive log-likelihood on FMNIST, CIFAR-10, and SVHN, evaluated across Average (Avg) and CVaR metrics at varying risk levels. Results are reported as mean $\pm$ standard deviation over five independent runs.

| Method | FMNIST | | | | CIFAR-10 | | | | SVHN | | | |
|---|---|---|---|---|---|---|---|---|---|---|---|---|
| | Avg | $CVaR_{0.5}$ | $CVaR_{0.7}$ | $CVaR_{0.9}$ | Avg | $CVaR_{0.5}$ | $CVaR_{0.7}$ | $CVaR_{0.9}$ | Avg | $CVaR_{0.5}$ | $CVaR_{0.7}$ | $CVaR_{0.9}$ |
| IWNPs | $2.81_{\pm0.02}$ | $2.31_{\pm0.02}$ | $1.78_{\pm0.05}$ | $-2.93_{\pm0.66}$ | $\underline{7.78}_{\pm0.07}$ | $\underline{7.29}_{\pm0.08}$ | $6.98_{\pm0.09}$ | $6.30_{\pm0.11}$ | $\underline{9.27}_{\pm0.03}$ | $8.06_{\pm0.03}$ | $7.08_{\pm0.03}$ | $4.99_{\pm0.04}$ |
| CVaR-NPs | $2.74_{\pm0.03}$ | $2.74_{\pm0.03}$ | $\underline{2.74}_{\pm0.03}$ | $\underline{2.74}_{\pm0.03}$ | $7.19_{\pm0.16}$ | $7.19_{\pm0.16}$ | $7.18_{\pm0.15}$ | $7.19_{\pm0.17}$ | $7.36_{\pm0.04}$ | $7.35_{\pm0.04}$ | $7.36_{\pm0.04}$ | $7.35_{\pm0.03}$ |
| GDRO-NPs | $2.75_{\pm0.02}$ | $\mathbf{2.75}_{\pm0.02}$ | $\mathbf{2.75}_{\pm0.02}$ | $\mathbf{2.75}_{\pm0.02}$ | $7.20_{\pm0.21}$ | $7.20_{\pm0.21}$ | $\underline{7.20}_{\pm0.21}$ | $\underline{7.21}_{\pm0.21}$ | $5.68_{\pm0.05}$ | $5.68_{\pm0.05}$ | $5.68_{\pm0.05}$ | $5.68_{\pm0.05}$ |
| TDRO-NPs | $\underline{2.83}_{\pm0.02}$ | $-6.67_{\pm1.62}$ | $-18.43_{\pm2.97}$ | $-33.96_{\pm3.51}$ | $7.63_{\pm0.07}$ | $6.16_{\pm0.09}$ | $5.20_{\pm0.11}$ | $3.22_{\pm0.15}$ | $9.14_{\pm0.03}$ | $\underline{8.56}_{\pm0.04}$ | $\underline{8.22}_{\pm0.03}$ | $\underline{7.55}_{\pm0.06}$ |
| OS-NPs(Ours) | $\mathbf{2.85}_{\pm0.02}$ | $\underline{2.75}_{\pm0.02}$ | $2.62_{\pm0.04}$ | $2.01_{\pm0.09}$ | $\mathbf{7.85}_{\pm0.09}$ | $\mathbf{7.60}_{\pm0.10}$ | $\mathbf{7.47}_{\pm0.11}$ | $\mathbf{7.22}_{\pm0.12}$ | $\mathbf{9.29}_{\pm0.03}$ | $\mathbf{8.74}_{\pm0.05}$ | $\mathbf{8.47}_{\pm0.04}$ | $\mathbf{7.95}_{\pm0.06}$ |

*Table 3.* Sim2Real Evaluation. All models meta-trained on the Lotka–Volterra are tested on the Hudson's Bay lynx–hare dataset.

| Method | Avg | $CVaR_{0.5}$ | $CVaR_{0.7}$ | $CVaR_{0.9}$ |
|---|---|---|---|---|
| IWNPs | $-78.00_{\pm1.05}$ | $-80.32_{\pm0.86}$ | $-80.72_{\pm0.96}$ | $-81.04_{\pm0.63}$ |
| CVaR-NPs | $\underline{-60.53}_{\pm0.64}$ | $\underline{-61.98}_{\pm1.14}$ | $\underline{-62.57}_{\pm0.91}$ | $\underline{-62.20}_{\pm0.36}$ |
| GDRO-NPs | $-63.62_{\pm0.96}$ | $-65.41_{\pm0.72}$ | $-65.83_{\pm0.87}$ | $-65.74_{\pm0.31}$ |
| TDRO-NPs | $-99.20_{\pm2.72}$ | $-102.31_{\pm1.70}$ | $-101.87_{\pm1.71}$ | $-101.70_{\pm1.64}$ |
| OS-NPs (Ours) | $\mathbf{-60.34}_{\pm0.18}$ | $\mathbf{-61.68}_{\pm0.26}$ | $\mathbf{-62.12}_{\pm0.08}$ | $\mathbf{-62.19}_{\pm0.27}$ |

rics and surpass IWNPs by over 23% in $CVaR_{0.9}$, demonstrating robust OOD performance. CVaR-NPs also exhibit strong generalization in OOD scenarios. More visualizations can be found in Appendix D.

## 4.4. Ablation Studies

We benchmark OS-NPs through an ablation study against other heuristics and assess its sensitivity to binning numbers.

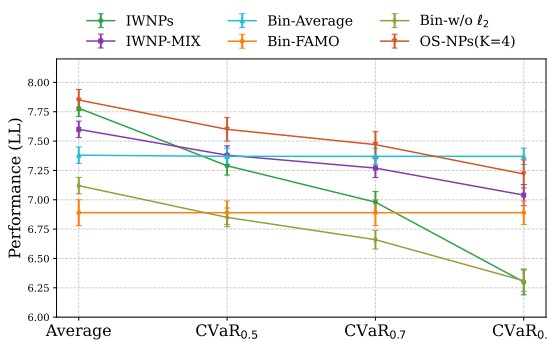

*Figure 6.* OS-NPs against Various Optimization Strategies.

**Comparison to Other Heuristic Strategies.** We evaluate OS-NPs against several heuristic strategies. IWNP-MIX is a linear combination of IWNPs and CVaR-NPs objectives with equal weights. Other strategies include stratum-based strategies: Bin-Average (arithmetic mean of stratum-wise losses), Bin-FAMO (Liu et al., 2023a) (multi-

objective optimization of losses), and Bin-w/o $\ell_2$ (without $\xi$-regularization). In Fig. 6, OS-NPs consistently illustrate superior performance and robustness in both average log-likelihood and tail-risk metrics ($CVaR_{0.5}$ to $CVaR_{0.9}$). Bin-Average encounters the prior collapse and sacrifices the average for worst-case improvement, significantly beating IWNPs in CVaR values. The performance gap between OS-NPs and Bin-w/o $\ell_2$, dropping to 6.31 in $CVaR_{0.9}$, examines the importance of $\ell_2$ regularization for generalization. Bin-FAMO's under-performance in average metrics suggests that simply using SoTA MTL methods hardly balance IWNPs' optimization without proper inductive bias.

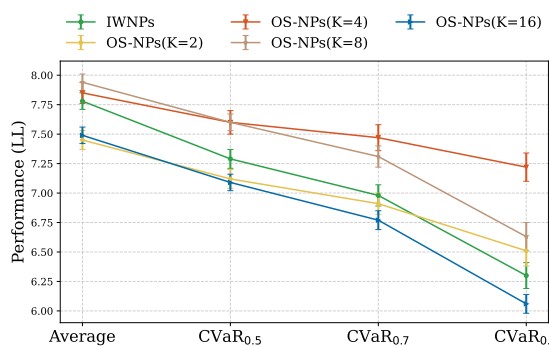

*Figure 7.* OS-NPs' Performance Sensitivity to Strata Number $K$.

**Impact of the Number of Strata $K$.** To evaluate the sensitivity to the binning number, we vary $K \in \{2, 4, 8, 16\}$, as summarized in Fig. 7. OS-NP ($K = 4$) emerges as the optimal configuration, while sufficiently smaller or larger $K$ leads to poor robustness performance. The results indicate a non-monotonic relationship between $K$ and performance, likely due to optimization instability or over-partitioning. Fortunately, OS-NP does not require significant tuning of $K$, as it sets $K = 4$ for all benchmarks.

## 5. Conclusion

This work identifies a robustness limitation of IWNPs in the latent space. We address the trade-off between average performance and tail risk generative robustness with OS-NPs.

The regularized worst-case formulation enables balanced gradient allocation across strata at minimal computational cost. Extensive experiments show consistent robustness improvements with competitive average performance.

## Acknowledgment

Dr. Qi Wang acknowledges support from the National Natural Science Foundation of China (NSFC) under the grant number 62306326.

## Impact Statement

This work introduces a robust optimization framework for Neural Processes by addressing gradient imbalances in latent space importance sampling. By employing an amortized gradient alignment strategy, the proposed method enhances the reliability of probabilistic meta-learning without the prohibitive computational costs of standard bi-level optimization. The approach operates primarily as an optimization refinement and an objective function modification, without introducing autonomous capabilities or altering the underlying generative architecture. Accordingly, it does not pose new societal or ethical risks beyond those inherent to probabilistic models, and its potential benefit lies in improving the safety and consistency of uncertainty estimation in risk-sensitive applications.

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

# A. Meta Learning and Neural Processes

**Meta Learning Methods.** Meta-learning has emerged as a dominant paradigm for rapid adaptation. Its core objective is to enable models to generalize to new but related tasks using only a handful of observations, by extracting and reusing transferable knowledge accumulated from prior tasks (Hospedales et al., 2021). Existing approaches can be broadly grouped into several categories. (i) Context-based approaches (Wang et al., 2026) typically adopt an encoder–decoder formulation, in which task-specific evidence from limited samples is compressed into latent representations. Representative examples include the NP line of work (Garnelo et al., 2018b; Wang et al., 2023; Gondal et al., 2021; Garnelo et al., 2018a), which seek to approximate exchangeable stochastic processes using neural architectures. (ii) Optimization-based methods frame meta-learning as a learning-to-optimize problem, allowing parameters to be efficiently adapted across tasks. A canonical example is Model-Agnostic Meta-Learning (MAML) (Finn et al., 2017; 2018; Abbas et al., 2022; Rajeswaran et al., 2019; Wang et al., 2025; Qu et al., 2025b; Wang et al., 2024a), which formulates adaptation as a bilevel optimization problem and has inspired numerous variants. (iii) Metric-based techniques (Snell et al., 2017; Allen et al., 2019) learn embedding spaces in which task similarity is explicitly encoded, making them particularly effective for few-shot visual recognition. Beyond these categories, alternative paradigms have also been explored, including hypernetwork-based formulations (Ha et al., 2016; Sendera et al., 2023) and recurrent meta-learning approaches (Duan et al., 2016), among others. However, this work mainly focuses on the NP family and studies the importance-weighted methods.

**Neural Process Family.** Existing NP research primarily addresses underfitting by enriching structural inductive biases or modifying inference objectives, e.g., importance-weighted or single-particle formulations. These efforts have led to a range of extensions that improve the expressive power and inferential capacity of standard NPs. On the architectural side, several works introduce inductive biases aligned with data symmetries. ConvCNP and ConvNP exploit translation equivariance via convolutional designs (Gordon et al., 2019; Foong et al., 2020), while transformer-based NPs encode invariance and equivariance through sequence modeling (Nguyen & Grover, 2022). To relax idealized symmetry assumptions, approximately equivariant NPs are proposed in (Ashman et al., 2024; Kawano et al., 2021; Holderrieth et al., 2021). Additional representations are explored through spectral-domain modeling (Mohseni & Duffield, 2024) and hierarchical tree-structured constructions (Tai & Guo, 2024). A complementary line of work focuses on improving inference and optimization. Attentive Neural Processes mitigate underfitting using local deterministic embeddings (Kim et al., 2019), while geometry-aware regularization (Venkataramanan & Denzler, 2025) and algorithmic stability analysis (Liu et al., 2023b) enhance uncertainty calibration and solution stability. Dutordoir et al. (2023) present a diffusion version of neural processes to improve the generation performance, and Mathieu et al. (2023) further incorporate symmetries into it for better generalization. Expressiveness of NPs can also be increased through autoregressive modeling (Bruinsma et al., 2023), hierarchical variational inference (Wang & Van Hoof, 2020), and mixture-of-expert meta-representations (Wang & Van Hoof, 2022). These developments enable application-oriented frameworks such as versatile NPs for high-dimensional implicit representations (Guo et al., 2023). In contrast to prior work, we study NP models from a robustness perspective, focusing on latent-space optimization, which remains relatively underexplored. Besides, OS-NPs are agnostic to architectural advances. The main paper uses a convolution neural architecture as the default, yet the appendix also runs experiments with MLP backbones. Our design also reserves the potential of combining with Transformer structures (Vaswani et al., 2017), paving the way for scalable robust optimization in large language models (Guo et al., 2025; Qu et al., 2025a; Mao et al., 2026; Qu et al., 2026; Zou et al., 2025a;d).

**Robust Optimization in Meta Learning.** Robust optimization in meta-learning often leverages worst-case objectives to navigate distributional shifts (Tay et al., 2022) or to mitigate performance degradation across poorly performing subgroups (Sagawa* et al., 2020). Building on this principle, Collins et al. (2020) incorporated worst-case optimization into the meta-learning framework by reformulating MAML to achieve task-robust initializations. Subsequent research has expanded this adversarial lens: Wang et al. (2021) enhanced adaptation via robust regularization, while Goldblum et al. (2020) introduced "Adversarial Querying" to protect few-shot learners against image-level perturbations. Further refining the objective, Wang et al. (2024b) and Rufolo et al. (2025) proposed distributionally robust frameworks that explicitly minimize worst-case fast adaptation risk. These robustness strategies have recently extended beyond simple perturbations to address structural and distributional heterogeneity. For instance, Sadeghi & Giannakis (2024) utilized adversarial adaptation to improve generalization under structural shifts in graph meta-learning, while Kong et al. (2020) employed a spectral approach to handle malicious outliers in the extreme "small-data" regime. Addressing the complexity of task sources, Zhang et al. (2023) modeled task distributions as hierarchical Gaussian mixtures to facilitate the detection of novel, unseen tasks through density estimation. Related stability–plasticity and interference-mitigation issues have also been studied in continual

learning, where biologically inspired structural priors and decorrelation mechanisms are used to improve stable adaptation across tasks (Zou et al., 2025b;c). Complementary to these density-based and adversarial approaches, our work investigates robustness through the lens of latent-space optimization within NP-based meta-learning models.

## B. Limitations and Novelty Clarification

**Limitation Discussion.** In balanced optimization, we introduce a regularized robust optimization problem and solve it in an approximate manner. The amortized approximation in Eq. (17) trades exactness for efficiency and presents a controlled bias proportional to the step size. Though we have demonstrated that the approximate error can be well controlled over iterations in Proposition C.2, the biased problem-solver can still restrict the optimization pipeline from attaining its best performance.

**Novelty Clarification.** It is worth noting that distinguished from some robust meta learning methods at the task level (Lv et al., 2024; Wang et al., 2024b; Collins et al., 2020) or adversarial noise (Wang et al., 2021), our approach introduces Generative Robustness in Definition 2.1 at the inference particle level. It leverages the unique properties of IWNPs, where the predictive distribution is induced by a randomly sampled particle. The key innovation lies not just in the stratification itself, but in the alignment of stratified order statistics with gradient aggregation, enforcing robustness in a fundamentally different and more nuanced way. In other words, the Generative Robustness specifies risk over inference-induced randomness, which is of great significance when implementing NPs in the real world. This is also absent in the field of robust probabilistic meta-learning.

Rather than heuristically blending concepts like stratification or MGDA, the optimization pipeline offers a computationally efficient and theoretically sound approach to balance both average and worst-case performance. The introduction of a regularized inner-loop problem with amortized optimization enhances practicality even further. Importantly, this work conducts a technical analysis of the convergence properties and approximate gaps in amortization and provides a guarantee for deployment.

## C. Theoretical Analysis

### C.1. Details in Matthew Effect of IWNPs

We begin by scrutinizing the optimization dynamics of the standard IWNPs. While IWNPs are widely used for uncertainty estimation, we show that their objective function possesses an inherent bias that leads to a "Matthew Effect," where the optimization process focuses disproportionately on a subset of samples.

**Proposition C.1** (Gradient Concentration in IWNPs). *Let $\ell_{(b)} = -\ln p_\theta(\mathcal{D}_\tau^T \mid z^{(b)})$ be the negative log-likelihood for particle $b$, and $\ell_{b_{min}} = \min_b \ell_{(b)}$ be the minimum loss. Define the gap $\Delta^{(b)} = \ell_{(b)} - \ell_{b_{min}} \geq 0$. The gradient of the IWNPs objective $\mathcal{L}_{IWNPs} = \ln[\frac{1}{B}\sum_b \exp(-\ell_{(b)})]$ regarding $\theta$ is a weighted average:*

$$\nabla_\theta \mathcal{L}_{IWNPs}(\theta, \phi) = -\nabla_\theta \ell_{b_{min}} - \sum_{b \neq b_{min}} \frac{\exp\left(-\Delta^{(b)}\right)}{1 + \sum_{b' \neq b_{min}} \exp\left(-\Delta^{(b')}\right)} \left[\nabla_\theta \Delta^{(b)}\right]. \tag{20}$$

*As the gaps $\Delta^{(b)} \to \infty$ for all $b \neq b_{min}$, the weights $w_{b \neq b_{min}}$ vanish at an exponential rate, causing the gradient to collapse to the single best particle: $\nabla_\theta \mathcal{L}_{IWNPs} \approx -\nabla_\theta \ell_{min}$.*

*Proof.* For a collection of per-particle negative log-likelihoods $\ell_{(b)} := -\ln p_\theta(\mathcal{D}_\tau^T | z^{(b)})$ with $z^{(b)} \sim p_\phi(z|\mathcal{D}_\tau^C)$, we denote the maximum one by $\ell_{b_{min}} = \min_{b \in \{1,\ldots,B\}} \ell_{(b)}$. Further, let the difference term be $\Delta^{(b)} = \ell_{(b)} - \ell_{b_{min}} \in \mathbb{R}_{\geq 0}$, and we represent the summation term by $S := \sum_{b=1}^B \exp\left(-\Delta^{(b)}\right) = 1 + \sum_{b \neq b_{min}} \exp\left(-\Delta^{(b)}\right)$. Hence, the importance-weighted optimization objective can be decomposed into the best particle log-likelihood and others:

$$\mathcal{L}_{IWNPs}(\theta, \phi) = \ln\left[\frac{1}{B}\sum_{b=1}^B \exp\left(-\ell_{(b)}\right)\right] = \ln\left[\frac{1}{B}\sum_{b=1}^B \exp\left(-\ell_{b_{min}}\right)\exp\left(\ell_{b_{min}} - \ell_{(b)}\right)\right] \tag{21a}$$

$$= -\ell_{b_{min}} + \ln\left[\sum_{b=1}^B \exp\left(-\Delta^{(b)}\right)\right] - \ln B. \tag{21b}$$

Naturally, we can estimate the gradient w.r.t. the decoder $\theta$ accordingly:

$$\nabla_\theta \mathcal{L}_{\text{IWNPs}}(\theta, \phi) = -\nabla_\theta \ell_{b_{\min}} + \frac{1}{S} \nabla_\theta \left[ \sum_{b=1}^{B} \exp\left(-\Delta^{(b)}\right) \right] \tag{22a}$$

$$= -\nabla_\theta \ell_{b_{\min}} - \sum_{b \neq b_{\min}} \frac{\exp\left(-\Delta^{(b)}\right)}{1 + \sum_{b' \neq b_{\min}} \exp\left(-\Delta^{(b')}\right)} \left[ \nabla_\theta \Delta^{(b)} \right]. \tag{22b}$$

For the weight of the particle gradient $\frac{\exp\left(-\Delta^{(b)}\right)}{1 + \sum_{b' \neq b_{\min}} \exp\left(-\Delta^{(b')}\right)}$, the denominator is bounded, and when $\Delta^{(b)}$ is sufficiently large, the weight vanishes to zero in practice. In an extreme scenario, when $\Delta^{(b)} \, \forall b \neq b_{\min}$ are in a large scale, all particles except the best one receive no feasible gradient due to the shrinking weights, i.e., $\nabla_\theta \mathcal{L}_{\text{IWNPs}}(\theta, \phi) \approx -\nabla_\theta \ell_{b_{\min}}$. $\qquad\square$

The exponential weighting mechanism identified in Proposition C.1 suggests that IWNPs effectively operate as a winner-take-all optimizer, ignoring the diversity of the latent distribution. This motivates the need for a more balanced objective that can trade off average and tail-risk performance without collapsing into a single-particle optimizer.

Proposition C.1 also applies to the analysis of general properties of log-sum-exp. The Matthew effect in IWNPs is not about technical novelty, but about diagnosing an overlooked optimization pathology in IWNPs and illuminating its persistence during training and its impact on tail adaptation performance.

**Naive Multi-Objective Scalarization as the Lower Bound of IWNPs.** The derived multi-objective formulation directly correlates with the IWNPs objective. By Jensen's inequality, the IWNP objective admits the following lower bound:

$$\mathcal{L}_{\text{IWNPs}} \geq -\sum_{k=1}^{K} \frac{1}{K} \mathcal{L}_k(\theta) \tag{23}$$

While this formulation implicitly adopts a uniform linear scalarization, such a static weighting scheme fails to account for the competitive nature of conflicting gradients across diverse tasks (Hu et al., 2023). To enhance robustness and navigate the trade-offs between heterogeneous stratum-wise losses, we transition from a fixed average to a dynamic, amortized optimization rule. This approach adaptively reweights objectives during training, ensuring that the model remains sensitive to underperforming components.

### C.2. Amortized Optimization Step-by-Step

The gradient of $\mathcal{J}(\xi)$ with respect to the log-parameters $\xi$ is given by:

$$\nabla_\xi \mathcal{J}(\xi) = \frac{\partial \alpha}{\partial \xi}^\top G^\top G \alpha + \lambda \xi, \tag{24}$$

Direct computation of Eq. (24) is computationally prohibitive for large-scale problems. The bottleneck stems from the explicit construction of the Gram matrix $G^\top G$ at each iteration: first, it requires $K$ individual backpropagation passes to compute the gradients $\{\nabla_\theta \mathcal{L}_k\}_{k=1}^{K}$, incurring a cost of $O(Kd)$ where $d$ is the parameter dimensionality; second, it necessitates $K^2$ pairwise inner product operations to populate the matrix, each requiring $O(d)$ multiplications, resulting in an additional $O(K^2 d)$ algebraic overhead. The total per-iteration complexity of $O((K^2 + K)d)$ becomes unsustainable as both the number of components $K$ and the dimensionality $d$ scale.

To ensure scalability, we adopt a time-amortized approximation that bypasses this explicit construction. Instead of treating the optimization problem as a static, isolated task at each step, our approach leverages the temporal continuity of the optimization trajectory. By utilizing a difference-based surrogate, we approximate the product $G^\top G \alpha$ through the decline changes in losses across successive updates. This strategy effectively amortizes the computational burden of the Gram matrix across the optimization trajectory, substituting an expensive, matrix-intensive operation with a sequence of efficient updates based on the observed loss decay rates.

Consider the model parameters $\theta$ at time step $t + 1$. The update rule follows:

$$\theta^{t+1} = \theta^t - \eta \sum_{k=1}^{K} \alpha_k^t \nabla_\theta \mathcal{L}_k^t, \tag{25}$$

where $\eta$ is the learning rate.

Let $\Delta\theta^{t+1} = \theta^{t+1} - \theta^t = -\eta\, G^t\alpha^t$ denote the parameter displacement. For a sufficiently small $\eta$, we apply a first-order Taylor expansion to the $k$-th stratum-wise loss $\mathcal{L}_k$ around $\theta^t$:

$$\mathcal{L}_k^{t+1} - \mathcal{L}_k^t \approx \langle \nabla_\theta \mathcal{L}_k^t, \Delta\theta^{t+1}\rangle. \tag{26}$$

Substituting the expression for $\Delta\theta^{t+1}$ yields:

$$\begin{aligned}
\mathcal{L}_k^{t+1} - \mathcal{L}_k^t &\approx -\eta\langle \nabla_\theta \mathcal{L}_k^t, G^t\alpha^t\rangle \\
&= -\eta\left[(G^t)^\top G^t\alpha^t\right]_k,
\end{aligned} \tag{27}$$

where $[\cdot]_k$ denotes the $k$-th element of the vector.

Rearranging this term provides an efficient estimator for the weighted gradient inner product:

$$\left[(G^t)^\top G^t\alpha^t\right]_k \approx \frac{\mathcal{L}_k^t - \mathcal{L}_k^{t+1}}{\eta} \triangleq \Delta\mathcal{L}_k^{t+1}. \tag{28}$$

Based on the difference-based surrogate derived in Eq. (28), the log-parameters $\xi$ are updated via gradient descent.

At time step $t+1$, given a learning rate $\beta$, the update rule for $\xi$ is formulated as:

$$\xi^{t+1} \leftarrow \xi^t - \beta\left[\frac{\partial\alpha^t}{\partial\xi}^\top \Delta\mathcal{L}^{t+1} + \lambda\xi^t\right], \tag{29}$$

Finally, the updated stratum-wise weights are obtained via $\alpha^{t+1} = \mathrm{softmax}(\xi^{t+1})$.

While the amortized update rule significantly reduces computational overhead, it introduces a first-order approximation to the true gradient projection. To ensure that this approximation does not compromise optimization integrity, we establish the following theoretical guarantees, demonstrating that the approximation error of the difference-based surrogate decreases quadratically with the step size.

**Proposition C.2** (Bounded Approximation Error). *Under Assumption 1, together with the condition $\max_k \|\nabla_\theta\mathcal{L}_k\| \leq F$, the discrepancy between the actual loss change and the task-specific gradient projection is bounded by:*

$$\left|\mathcal{L}_i^{t+1} - \mathcal{L}_i^t - \eta(\nabla_\theta\mathcal{L}_i^t)^\top d^t\right| \leq \mathcal{O}(\eta^2).$$

*Proof.* To bound the approximation error, we first establish that the descent direction $d^t$ inherits the bound from individual task gradients. Since $d^t = \sum_{j=1}^k \alpha_j\nabla\mathcal{L}_j^t$ and the weight vector $\alpha$ lies in the probability simplex $\Delta^k$, the triangle inequality and the gradient bound assumption $\|\nabla\mathcal{L}_i\| \leq F$ imply:

$$\|d^t\| = \left\|\sum_{j=1}^k \alpha_j\nabla\mathcal{L}_j^t\right\| \leq \sum_{j=1}^k \alpha_j\|\nabla\mathcal{L}_j^t\| \leq \sum_{j=1}^k \alpha_j F = F. \tag{30}$$

Consequently, the update magnitude is strictly controlled by the learning rate as $\|\Delta\theta_t\| = \|\eta d^t\| \leq \eta F$.

Applying the Fundamental Theorem of Calculus, the exact difference in the $i$-th task loss is given by the integral of the gradient along the parameter update path:

$$\mathcal{L}_i^{t+1} - \mathcal{L}_i^t = \int_0^1 \nabla\mathcal{L}_i(\theta_t + \tau\Delta\theta_t)^T \Delta\theta_t\, d\tau. \tag{31}$$

By subtracting the first-order approximation term $\nabla\mathcal{L}_i^{tT}\Delta\theta_t = \int_0^1 \nabla\mathcal{L}_i^{tT}\Delta\theta_t\, d\tau$ from both sides and substituting $\Delta\theta_t = \eta d^t$, we express the approximation residual as:

$$\mathcal{L}_i^{t+1} - \mathcal{L}_i^t - \eta\nabla\mathcal{L}_i^{tT}d^t = \int_0^1 (\nabla\mathcal{L}_i(\theta_t + \tau\eta d^t) - \nabla\mathcal{L}_i^t)^T(\eta d^t)\, d\tau. \tag{32}$$

Utilizing the Cauchy-Schwarz inequality and the $L$-Lipschitz continuity of the gradient $\nabla \mathcal{L}_i$, the absolute value of the integrand is upper-bounded by:

$$|(\nabla \mathcal{L}_i(\theta_t + \tau \eta d^t) - \nabla \mathcal{L}_i^t)^T (\eta d^t)| \leq \|\nabla \mathcal{L}_i(\theta_t + \tau \eta d^t) - \nabla \mathcal{L}_i^t\| \cdot \|\eta d^t\| \tag{33}$$

$$\leq (L\tau\eta\|d^t\|) \cdot (\eta\|d^t\|) = L\tau\eta^2\|d^t\|^2. \tag{34}$$

Integrating this bound with respect to $\tau$ over the interval $[0,1]$ yields:

$$|\mathcal{L}_i^{t+1} - \mathcal{L}_i^t - \eta \nabla \mathcal{L}_i^{tT} d^t| \leq L\eta^2\|d^t\|^2 \int_0^1 \tau d\tau = \frac{L\|d^t\|^2}{2}\eta^2. \tag{35}$$

Finally, substituting the gradient norm bound $\|d^t\| \leq F$ completes the proof, demonstrating that the finite-difference approximation error is indeed $\mathcal{O}(\eta^2)$. $\qquad\square$

### C.3. Regularization and Convergence Analysis

In this section, we provide the formal statements and detailed proofs for the theoretical properties of our proposed method. We first establish the well-posedness of the optimization objective through the existence and uniqueness of the optimal weight assignment, followed by a convergence analysis of OS-NPs. Finally, we characterize the gradient balancing property at the stationary points.

**Theorem C.3** (Existence and Uniqueness). *Suppose the task gradients are bounded, i.e., $\max_k \|\nabla_\theta \mathcal{L}_k\|_2 \leq F$. For any $\lambda > 2F^2$, $\mathcal{J}(\xi)$ is strictly strongly convex over $\mathbb{R}^K$. Consequently, the optimal solution $\xi^*$ exists and is unique, yielding a well-defined weight assignment $\alpha^* = softmax(\xi^*)$.*

*Proof.* To establish the strong convexity of $\mathcal{J}(\xi)$, we analyze its Hessian $\nabla^2 \mathcal{J}(\xi)$. Let $G = [g_1, \ldots, g_K] \in \mathbb{R}^{d \times K}$ denote the matrix of basis vectors and $d = G\alpha = \sum_{k=1}^K \alpha_k g_k$ be the weighted sum. By applying the chain rule, the Hessian is given by:

$$\nabla^2 \mathcal{J}(\xi) = \mathbf{J}_\alpha^\top G^\top G \mathbf{J}_\alpha + \sum_{k=1}^K (d^\top g_k)\nabla_\xi^2 \alpha_k + \lambda \mathbf{I}, \tag{36}$$

where $\mathbf{J}_\alpha = \text{diag}(\alpha) - \alpha\alpha^\top$ is the Jacobian of the softmax mapping.

Since $G^\top G \succeq 0$, the first term $\mathbf{J}_\alpha^\top G^\top G \mathbf{J}_\alpha$ is positive semi-definite (PSD). The Hessian of the $k$-th softmax component $\alpha_k$ is:

$$\nabla_\xi^2 \alpha_k = \alpha_k \left[ (e_k - \alpha)(e_k - \alpha)^\top - \mathbf{J}_\alpha \right], \tag{37}$$

where $e_k \in \mathbb{R}^K$ is the $k$-th standard basis vector. For any unit vector $v \in \mathbb{R}^K$ ($\|v\|_2 = 1$), the quadratic form is:

$$v^\top (\nabla_\xi^2 \alpha_k) v = \alpha_k \left[ (v_k - \bar{v})^2 - \text{Var}_\alpha(v) \right], \tag{38}$$

where $\bar{v} = \sum_{j=1}^K \alpha_j v_j$ and $\text{Var}_\alpha(v) = \sum_{j=1}^K \alpha_j (v_j - \bar{v})^2$ denote the weighted mean and variance of $v$ under the distribution $\alpha$.

For any unit vector $v$, we have

$$|v^\top (\nabla_\xi^2 \alpha_k) v| = \alpha_k \left| (v_k - \bar{v})^2 - \text{Var}_\alpha(v) \right|.$$

Using the inequality $(v_k - \bar{v})^2 \leq 2(v_k^2 + \bar{v}^2)$ and the fact that $\text{Var}_\alpha(v) \geq 0$, we obtain

$$|v^\top (\nabla_\xi^2 \alpha_k) v| \leq \alpha_k \left( (v_k - \bar{v})^2 + \text{Var}_\alpha(v) \right).$$

Observe that $\sum_{k=1}^K \alpha_k (v_k - \bar{v})^2 = \text{Var}_\alpha(v)$ and $\sum_{k=1}^K \alpha_k = 1$. Therefore,

$$\sum_{k=1}^K |v^\top (\nabla_\xi^2 \alpha_k) v| \leq 2\text{Var}_\alpha(v) \leq 2\max_k v_k^2 \leq 2.$$

Then $\|\nabla_\xi^2 \alpha_k\|_2 \leq 2$.

Under the assumption $\|g_k\|_2 \leq F$, we have $|d^\top g_k| \leq F^2$. Consequently, the minimum eigenvalue of the Hessian $\nabla^2 \mathcal{J}$ can be bounded as:

$$\lambda_{\min}(\nabla^2 \mathcal{J}) \geq \lambda + \lambda_{\min}\left(\sum_{k=1}^K (d^\top g_k)\nabla_\xi^2 \alpha_k\right) \tag{39}$$

$$\geq \lambda - 2F^2. \tag{40}$$

For $\lambda > 2F^2$, it follows that $\nabla^2 \mathcal{J} \succ 0$, rendering $\mathcal{J}(\xi)$ strictly strongly convex. This ensures the existence and uniqueness of the minimizer $\xi^*$, which further implies the uniqueness of $\alpha^* = \text{softmax}(\xi^*)$. $\qquad\square$

*Remark* C.4. While we may conservatively set a small $\lambda$ in practice, the condition $\lambda > 2F^2$ is increasingly likely to be satisfied as training progresses and the gradient norms $\|\nabla_\theta \mathcal{L}_k\|_2$ diminish. Consequently, our regularization ensures that the weight assignment problem becomes increasingly well-posed in the later stages of optimization, providing the necessary curvature to stabilize convergence even if the condition is temporarily violated during the initial high-gradient phase.

Regarding the convergence of OS-NPs, our analysis addresses a key departure from the vanilla Robbins-Monro (RM) framework. While RM entails a single-level process, OS-NPs involve a nested, two-level iteration. Because the inner loop is solved only approximately, the resulting hyper-gradient estimates are inherently biased, precluding the direct application of standard RM theory. Consequently, we establish a conservative yet rigorous guarantee that the limit inferior of the gradient norm vanishes asymptotically:

**Theorem C.5** (Convergence of Stratified Gradient Descent). *Under Assumptions 1/2 and step-size conditions $\sum_{t=1}^\infty \eta^t = \infty, \sum_{t=1}^\infty (\eta^t)^2 < \infty, \sum_{t=1}^\infty \beta^t < \infty$, Algorithm (1) guarantees $\liminf_{t\to\infty} \|d^t\|^2 = 0$ in Eq. (18), guaranteeing convergence to a stationary point.*

*Proof.* Since each $\mathcal{L}_k$ is $L$-smooth, we invoke the standard descent lemma:

$$\mathcal{L}_k^{t+1} \leq \mathcal{L}_k^t + \langle \nabla \mathcal{L}_k^t, \theta^{t+1} - \theta^t \rangle + \frac{L}{2}\|\theta^{t+1} - \theta^t\|^2. \tag{41}$$

Substituting the update rule $\theta^{t+1} - \theta^t = -\eta^t d^t$, we obtain:

$$\mathcal{L}_k^{t+1} \leq \mathcal{L}_k^t - \eta^t \langle \nabla \mathcal{L}_k^t, d^t \rangle + \frac{L(\eta^t)^2}{2}\|d^t\|^2. \tag{42}$$

Let $\Phi_t^t = \sum_{k=1}^K \alpha_k^t \mathcal{L}_k^t$. Since the parameter iterates $\{\theta^t\}$ are contained within the compact set $\Theta$ and the loss functions are continuous, it follows that $\Phi_t^t$ is uniformly bounded. Moreover, let $d^t = \sum_{k=1}^K \alpha_k^t \nabla_\theta \mathcal{L}_k^t$; as $\nabla_\theta \mathcal{L}_k$ is continuous under Assumption 1 and the parameters remain in the compact set $\Theta$, the update direction $d^t$ is also uniformly bounded. Multiplying the above inequality by $\alpha_k^t$ and summing over $k$:

$$\Phi_t^{t+1} \leq \Phi_t^t - \eta^t \|d^t\|^2 + \frac{L(\eta^t)^2}{2}\|d^t\|^2 = \Phi_t^t - \eta^t(1 - \frac{L\eta^t}{2})\|d^t\|^2, \tag{43}$$

To account for the update in weights $\alpha$, we observe:

$$\Phi_{t+1}^{t+1} = \Phi_t^{t+1} + \sum_{k=1}^K \left(\alpha_k^{t+1} - \alpha_k^t\right)\mathcal{L}_k^{t+1}. \tag{44}$$

Given Assumptions 1/2 and the Softmax function is $L_s$-Lipschitz continuous, the change in weights is bounded by the update of the auxiliary variables $\xi$:

$$\|\Phi_{t+1}^{t+1} - \Phi_t^{t+1}\| \leq M\sum_{k=1}^K \|\alpha_k^{t+1} - \alpha_k^t\| \leq ML_s\sum_{k=1}^K \|\xi_k^{t+1} - \xi_k^t\| \leq O(\beta^t). \tag{45}$$

Combining Eq. (43) and (45), we have:

$$\Phi_{t+1}^{t+1} \leq \Phi_t^t - \eta^t\left(1 - \frac{L\eta^t}{2}\right)\|d^t\|^2 + O(\beta^t). \tag{46}$$

Summing from $t = 1$ to $T$ yields:

$$\Phi_{T+1}^{T+1} \leq \Phi_1^1 - \sum_{t=1}^{T} \eta^t \left(1 - \frac{L\eta^t}{2}\right) \|d^t\|^2 + O\left(\sum_{t=1}^{T} \beta^t\right). \tag{47}$$

Rearranging Eq. (47), we obtain:

$$\sum_{t=1}^{T} \eta^t \|d^t\|^2 \leq \Phi_1^1 - \Phi_{T+1}^{T+1} + \frac{L}{2} \sum_{t=1}^{T} (\eta^t)^2 \|d^t\|^2 + O\left(\sum_{t=1}^{T} \beta^t\right). \tag{48}$$

Since $\Phi_{T+1}^{T+1}$ and $\|d^t\|$ are uniformly bounded for all $T$ and $t$ respectively, and given the summability conditions $\sum_{t=1}^{\infty} (\eta^t)^2 < \infty$ and $\sum_{t=1}^{\infty} \beta^t < \infty$, the right-hand side of Eq. (48) remains upper-bounded by a constant independent of $T$ as $T \to \infty$.

If $\liminf_{t\to\infty} \|d^t\|^2 > 0$, there must exist some $\epsilon > 0$ and a time $t_0$ such that $\|d^t\|^2 \geq \epsilon$ for all $t > t_0$. Given the condition $\sum_{t=1}^{\infty} \eta^t = \infty$, it follows that the series $\sum_{t=1}^{\infty} \eta^t \|d^t\|^2$ would diverge to infinity. This directly contradicts the previously established upper bound on the right-hand side of Eq. (48), thereby forcing $\liminf_{t\to\infty} \|d^t\|^2 = 0$. $\qquad \square$

Proposition C.6 establishes that the objective in Eq. (16) prevents the optimization weights from collapsing to the vertices of the simplex, thereby ensuring they do not converge to 0 or 1.

**Proposition C.6** (Gradient Balancing). *Under assumption 1. For any first-order stationary point $\xi^*$ of Eq. (16), we define the corresponding aggregated update direction as*

$$d^* = \sum_{k=1}^{K} \alpha_k^* \nabla_\theta \mathcal{L}_k, \tag{49}$$

*where $\alpha^* = \mathrm{softmax}(\xi^*)$. Then the following holds:*

$$\langle \nabla_\theta \mathcal{L}_k, d^* \rangle = \|d^*\|_2^2 - \frac{2\lambda \xi_k^*}{\alpha_k^*}, \ k \in \{1, \ldots, K\}. \tag{50}$$

*Moreover, at any stationary point, the corresponding weights $\alpha_k^*$ remain strictly interior to the simplex and cannot converge to 0 or 1.*

*Proof.* At a first-order stationary point, the partial derivative with respect to each log-weight $\xi_k$ vanishes, i.e., $\partial J / \partial \xi_k = 0$. Applying the chain rule together with the Softmax derivative $\frac{\partial \alpha_j}{\partial \xi_k} = \alpha_k(\delta_{kj} - \alpha_j)$, we obtain

$$\frac{\partial J}{\partial \xi_k} = \left(\sum_{j=1}^{K} \frac{\partial \alpha_j}{\partial \xi_k} \nabla_\theta \mathcal{L}_j\right)^\top d^* + 2\lambda \xi_k \tag{51a}$$

$$= \alpha_k \left(\nabla_\theta \mathcal{L}_k - \sum_{j=1}^{K} \alpha_j \nabla_\theta \mathcal{L}_j\right)^\top d^* + 2\lambda \xi_k = 0. \tag{51b}$$

Rearranging terms yields $\alpha_k \left(\langle \nabla_\theta \mathcal{L}_k, d^* \rangle - \|d^*\|_2^2\right) = -2\lambda \xi_k$. Since $\alpha_k > 0$ under the Softmax parameterization, dividing both sides by $\alpha_k$ gives Eq. (50).

We now show that no stationary point can lie on the boundary of the simplex. Suppose, for contradiction, that $\alpha_i \to 1$ for some index $i$. From the Softmax parameterization

$$\alpha_i = \frac{1}{1 + \sum_{j \neq i} \exp(\xi_j - \xi_i)}, \tag{52}$$

this limit implies $\xi_j - \xi_i \to -\infty$ for all $j \neq i$. Meanwhile, applying the stationarity condition to index $i$ yields

$$\alpha_i \left(\langle \nabla_\theta \mathcal{L}_i, d^* \rangle - \|d^*\|_2^2\right) = -2\lambda \xi_i. \tag{53}$$

As $\alpha_i \to 1$, the aggregated direction $d^*$ converges to $\nabla_\theta \mathcal{L}_i$, implying that the left-hand side approaches zero. For any fixed $\lambda > 0$, this requires $\xi_i \to 0$. Consequently, $\xi_j \to -\infty$ for all $j \neq i$. However, applying the stationarity condition to any $j \neq i$ gives

$$\alpha_j \left( \langle \nabla_\theta \mathcal{L}_j, d^* \rangle - \|d^*\|_2^2 \right) = -2\lambda \xi_j . \tag{54}$$

In this limit, the right-hand side diverges to $+\infty$ as $\xi_j \to -\infty$. Under Assumption 1, the gradient $\nabla_\theta \mathcal{L}_k$ is continuous over $\Theta$. Given that the update parameters reside within a compact space, the gradients $\nabla_\theta \mathcal{L}_k$ are uniformly bounded for all $k \in \{1, \ldots, K\}$. Consequently, the left-hand side vanishes as $\alpha_j \to 0$, leading to a direct contradiction. Therefore, $\alpha_i \to 1$ cannot occur at a stationary point. A symmetric argument rules out the case $\alpha_i \to 0$. In this regime, there must exist some $j$ such that $\xi_j - \xi_i \to +\infty$, implying $\xi_j \to +\infty$. Yet the stationarity condition for index $j$ would then require a bounded left-hand side to balance an unbounded term $-2\lambda \xi_j$, which is again impossible. We conclude that, under the regularized objective, all stationary solutions correspond to strictly interior weight vectors. This Softmax-regularized formulation therefore precludes degenerate winner-take-all behavior and enforces balanced gradient contributions across strata. $\qquad \square$

### C.4. Theoretical Analysis of the TDRO-NPs Objective

In the experiments, we involve TDRO (Gladin et al., 2025) to instantiate NPs as the baseline. Here, we perform analysis with respect to this baseline. The smoothed objective for TDRO-NPs is defined by the following log-sum-exp formulation:

$$\mathcal{J}_\lambda(\theta) = \lambda \ln \left( \frac{1}{B} \sum_{b=1}^{B} \exp \frac{\ell_{(b)}(\theta)}{\lambda} \right), \tag{55}$$

where $\ell_{(b)}(\theta)$ denotes the log-likelihood of the $b$-th latent particle. This formulation provides a unified framework that interpolates between different optimization behaviors via the temperature parameter $\lambda$. Notably, for $\lambda = 1$, Eq. (55) recovers the standard objective of IWNPs. As demonstrated below, the parameter $\lambda$ allows the model to span a spectrum from average-case to best-case optimization.

As $\lambda$ approaches zero, let $\ell_{(k)}(\theta) = \max_b \ell_{(b)}(\theta)$ be the highest log-likelihood among all particles. For any $j \neq k$, we have:

$$\ln \left( \frac{\exp(\ell_{(k)}(\theta)/\lambda)}{\exp(\ell_{(j)}(\theta)/\lambda)} \right) = \frac{\ell_{(k)}(\theta) - \ell_{(j)}(\theta)}{\lambda} \xrightarrow{\lambda \to 0^+} +\infty. \tag{56}$$

Consequently, the objective $\mathcal{J}_\lambda(\theta)$ converges to the hard-maximum:

$$\lim_{\lambda \to 0^+} \mathcal{J}_\lambda(\theta) = \ell_{(k)}(\theta) - \lim_{\lambda \to 0^+} \lambda \ln B = \max_b \ell_{(b)}(\theta). \tag{57}$$

In this regime, the model focuses exclusively on the most successful particle, exhibiting "best-case" optimization.

For large $\lambda$, we employ a second-order Taylor expansion of the exponential term:

$$\exp \left( \frac{\ell_{(b)}(\theta)}{\lambda} \right) \approx 1 + \frac{\ell_{(b)}(\theta)}{\lambda} + \frac{\ell_{(b)}(\theta)^2}{2\lambda^2}. \tag{58}$$

Summing over all particles and normalizing by $B$:

$$\ln \left( \frac{1}{B} \sum_{b=1}^{B} \exp \frac{\ell_{(b)}(\theta)}{\lambda} \right) \approx \ln \left( 1 + \frac{1}{\lambda B} \sum_{b=1}^{B} \ell_{(b)}(\theta) + \frac{1}{2\lambda^2 B} \sum_{b=1}^{B} \ell_{(b)}(\theta)^2 \right). \tag{59}$$

Applying the logarithmic approximation $\ln(1 + x) \approx x - \frac{x^2}{2}$ for small $x$:

$$\mathcal{J}_\lambda(\theta) \approx \lambda \left[ \frac{\sum_{b=1}^{B} \ell_{(b)}(\theta)}{\lambda B} + \frac{\sum_{b=1}^{B} \ell_{(b)}(\theta)^2}{2\lambda^2 B} - \frac{1}{2} \left( \frac{\sum_{b=1}^{B} \ell_{(b)}(\theta)}{\lambda B} \right)^2 \right] = \frac{1}{B} \sum_{b=1}^{B} \ell_{(b)}(\theta) + \frac{1}{2\lambda} \mathrm{Var}[\ell_{(b)}(\theta)]. \tag{60}$$

As $\lambda \to +\infty$, the variance term vanishes, and the objective converges to the empirical mean:

$$\lim_{\lambda \to +\infty} \mathcal{J}_\lambda(\theta) = \frac{1}{B} \sum_{b=1}^{B} \ell_{(b)}(\theta). \tag{61}$$

In this limit, the problem reduces to standard average-case optimization.

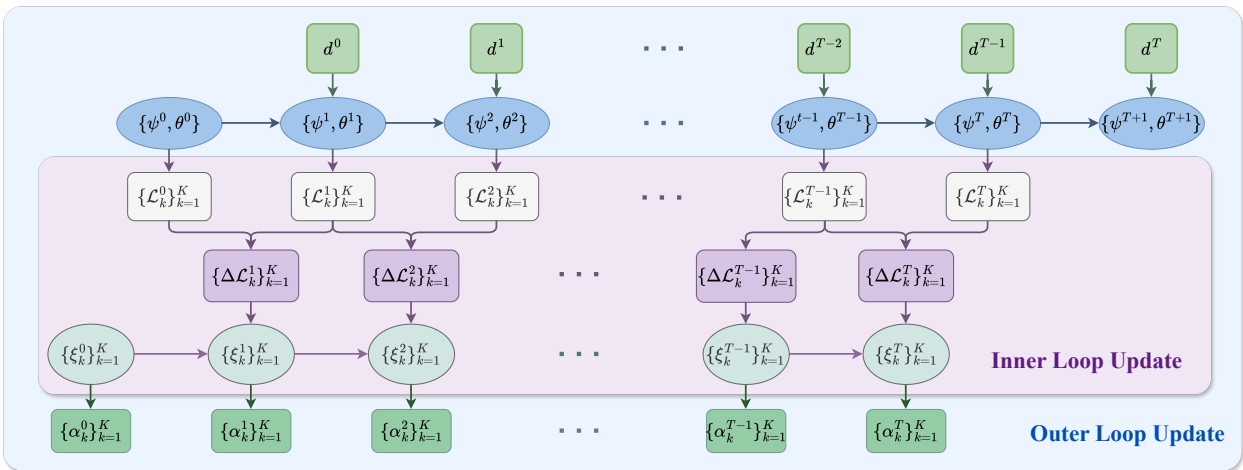

*Figure 8.* The Stratified Order-Statistic Optimization workflow proceeds from $t = 0 \rightarrow T$, beginning with the initialization of parameters $\{\psi^0, \theta^0\}$. At $t = 0$, $B$ particles are partitioned into $K$ strata to compute initial losses $\{\mathcal{L}_k^0\}_{k=1}^K$, which are aggregated via pre-specified weights to produce the first parameter update. In each subsequent iteration, an Inner Loop Update calculates stratum-wise loss improvements $\{\Delta\mathcal{L}_k^t\}_{k=1}^K$ to refine logits $\{\xi_k^t\}_{k=1}^K$, which are transformed via softmax into adaptive weights $\{\alpha_k^t\}_{k=1}^K$. Finally, the Outer Loop Update applies these weights to the loss gradients to perform weighted gradient descent, yielding the updated global parameters $\{\psi^{t+1}, \theta^{t+1}\}$.

## D. Additional Experimental Results

### D.1. Additional Results on mini-ImageNet

To further evaluate scalability on higher-complexity benchmarks, we conduct additional experiments on mini-ImageNet using the same protocols and hyperparameter settings as CIFAR-10 and SVHN. Table. 4 reports the meta-testing completion log-likelihoods under different CVaR levels, while Table. 5 summarizes the corresponding training time per epoch.

*Table 4.* Meta-testing completion log-likelihoods on mini-ImageNet.

| Methods | Average | $\text{CVaR}_{0.5}$ | $\text{CVaR}_{0.3}$ | $\text{CVaR}_{0.1}$ |
|---------|---------|---------|---------|---------|
| IWNPs | 5.519 | 4.135 | 3.449 | 2.275 |
| CVaR-NPs | 6.404 | 6.406 | 6.402 | 6.400 |
| GDRO-NPs | 2.120 | 2.119 | 2.119 | 2.118 |
| TDRO-NPs | 3.912 | 3.602 | 3.416 | 3.000 |
| OS-NPs | **6.895** | **6.678** | **6.559** | **6.278** |

*Table 5.* Training time per epoch on mini-ImageNet.

| Methods | Training Time (mins/epoch) |
|---------|---------|
| IWNPs | 19.51 |
| CVaR-NPs | 33.42 |
| GDRO-NPs | 20.33 |
| TDRO-NPs | 19.50 |
| OS-NPs | 21.08 |

The results show that OS-NPs consistently maintain strong average and tail-risk performance on mini-ImageNet with only marginal additional training overhead. Although CVaR-NPs achieve competitive robustness on mini-ImageNet under class-disjoint OOD generalization, they require substantially higher training cost than OS-NPs.

We further verify that OS-NPs maintain competitive best-particle performance, indicating that improvements in worst-case robustness do not degrade best-case predictions. Qualitative results are shown in Fig. 9.

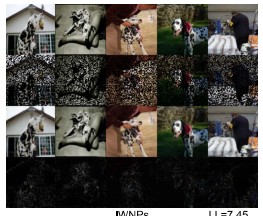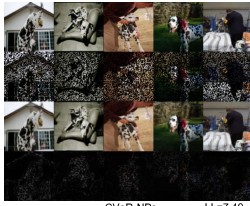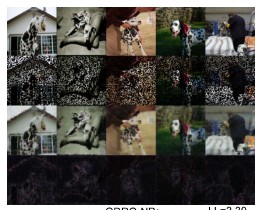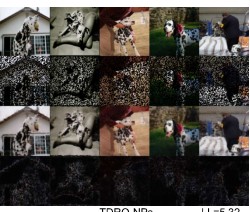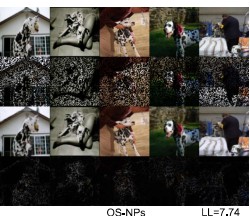

*Figure 9.* Best-particle generation results on mini-ImageNet. OS-NPs maintain high-quality reconstructions from top-performing particles with improved structural fidelity compared to baselines.

**Qualitative Sim-to-Real Results.** Figure. 11 presents additional qualitative results for Lynx–Hare population interpolation across IWNPs, CVaR-NPs, GDRO-NPs, TDRO-NPs, and OS-NPs. Notably, OS-NPs outperform the baseline models in tracking the population dynamics of the dataset.

**Sensitivity to $\lambda$ in the Inner Loop.** To evaluate the impact of regularization, we vary $\lambda \in \{0, 0.001, 0.01, 0.1\}$, as illustrated in Fig. 12. Theoretically, the $\ell_2$ term ensures optimization stability by enforcing strict strong convexity on $\mathcal{J}(\xi)$. Empirically, negligible values ($\lambda \leq 0.001$) yield suboptimal performance (Avg. LL 7.12) due to unstable latent weighting. Performance peaks within the $\lambda \in [0.01, 0.1]$ range, where $\lambda = 0.01$ maximizes average performance (7.85) and $\lambda = 0.1$ optimizes tail-risk robustness (CVaR$_{0.9}$=7.27). These results confirm that $\lambda$ under a controlled interval is not sensitive to balancing optimization stability with robust weighting across the risk landscape.

*Table 6.* Average training time per epoch in CIFAR-10 Image Completion (min).

| Method | IWNPs | CVaR-NPs | GDRO-NPs | TDRO-NPs | OS-NPs |
|---|---|---|---|---|---|
| Time (min) | **5.43** | 9.44 | 6.10 | 6.06 | 6.27 |

Table. 6 reports the average training duration per epoch for the CIFAR-10 image completion task. While the baseline IWNPs naturally achieve the fastest runtime at 5.43 minutes, our proposed OS-NPs exhibit competitive efficiency at 6.27 minutes. Notably, OS-NPs achieve superior robustness with only a marginal computational overhead compared to GDRO-NPs (6.10 min) and TDRO-NPs (6.06 min), while significantly outperforming CVaR-NPs (9.44 min) in terms of training speed.

Beyond the supplementary experiments on MNIST using convolutional models in Table. 7, we further evaluate our approach using Multi-Layer Perceptron (MLP) architectures to verify its structural robustness. As shown in Tables. 8 and 9, by reporting performance across a diverse range of datasets including MNIST, Fashion-MNIST, CIFAR-10, and SVHN, we empirically demonstrate that our findings hold without loss of generality across varied network structures.

*Table 7.* Predictive log-likelihood on MNIST (Conv architectures).

| Method | Avg | CVaR$_{0.5}$ | CVaR$_{0.7}$ | CVaR$_{0.9}$ |
|---|---|---|---|---|
| IWNPs | $3.01_{\pm 0.01}$ | $2.43_{\pm 0.04}$ | $0.88_{\pm 0.1}$ | $-14.68_{\pm 1.26}$ |
| CVaR-NPs | $2.82_{\pm 0.02}$ | $2.81_{\pm 0.02}$ | $2.82_{\pm 0.02}$ | $2.82_{\pm 0.02}$ |
| GDRO-NPs | $2.94_{\pm 0.02}$ | $2.93_{\pm 0.02}$ | **$2.94_{\pm 0.02}$** | **$2.94_{\pm 0.02}$** |
| TDRO-NPs | **$3.22_{\pm 0.01}$** | $1.59_{\pm 0.10}$ | $0.41_{\pm 0.20}$ | $-2.28_{\pm 0.30}$ |
| OS-NPs (Ours) | $3.16_{\pm 0.01}$ | **$3.02_{\pm 0.02}$** | $2.79_{\pm 0.02}$ | $1.33_{\pm 0.11}$ |

*Table 8.* Predictive log-likelihood on MNIST (MLP architectures).

| Method | Avg | CVaR$_{0.5}$ | CVaR$_{0.7}$ | CVaR$_{0.9}$ |
|---|---|---|---|---|
| IWNPs | $1.03_{\pm 0.001}$ | $0.87_{\pm 0.002}$ | $0.78_{\pm 0.003}$ | $0.62_{\pm 0.006}$ |
| CVaR-NPs | $1.01_{\pm 0.002}$ | **$1.01_{\pm 0.002}$** | $1.00_{\pm 0.002}$ | **$1.00_{\pm 0.002}$** |
| GDRO-NPs | $0.99_{\pm 0.002}$ | $0.99_{\pm 0.002}$ | $0.99_{\pm 0.002}$ | $0.99_{\pm 0.002}$ |
| TDRO-NPs | **$1.06_{\pm 0.001}$** | $0.95_{\pm 0.003}$ | $0.89_{\pm 0.003}$ | $0.78_{\pm 0.005}$ |
| OS-NPs (Ours) | $1.02_{\pm 0.001}$ | **$1.01_{\pm 0.001}$** | **$1.01_{\pm 0.001}$** | **$1.00_{\pm 0.002}$** |

# E. Implementation Details

Experimental implementations were developed using the PyTorch framework, with Adam serving as the primary stochastic optimization algorithm for all parameter updates. Fig. 8 illustrates the operational flow of the Stratified Order-Statistic Optimization framework throughout the duration $t \in [0, T]$.

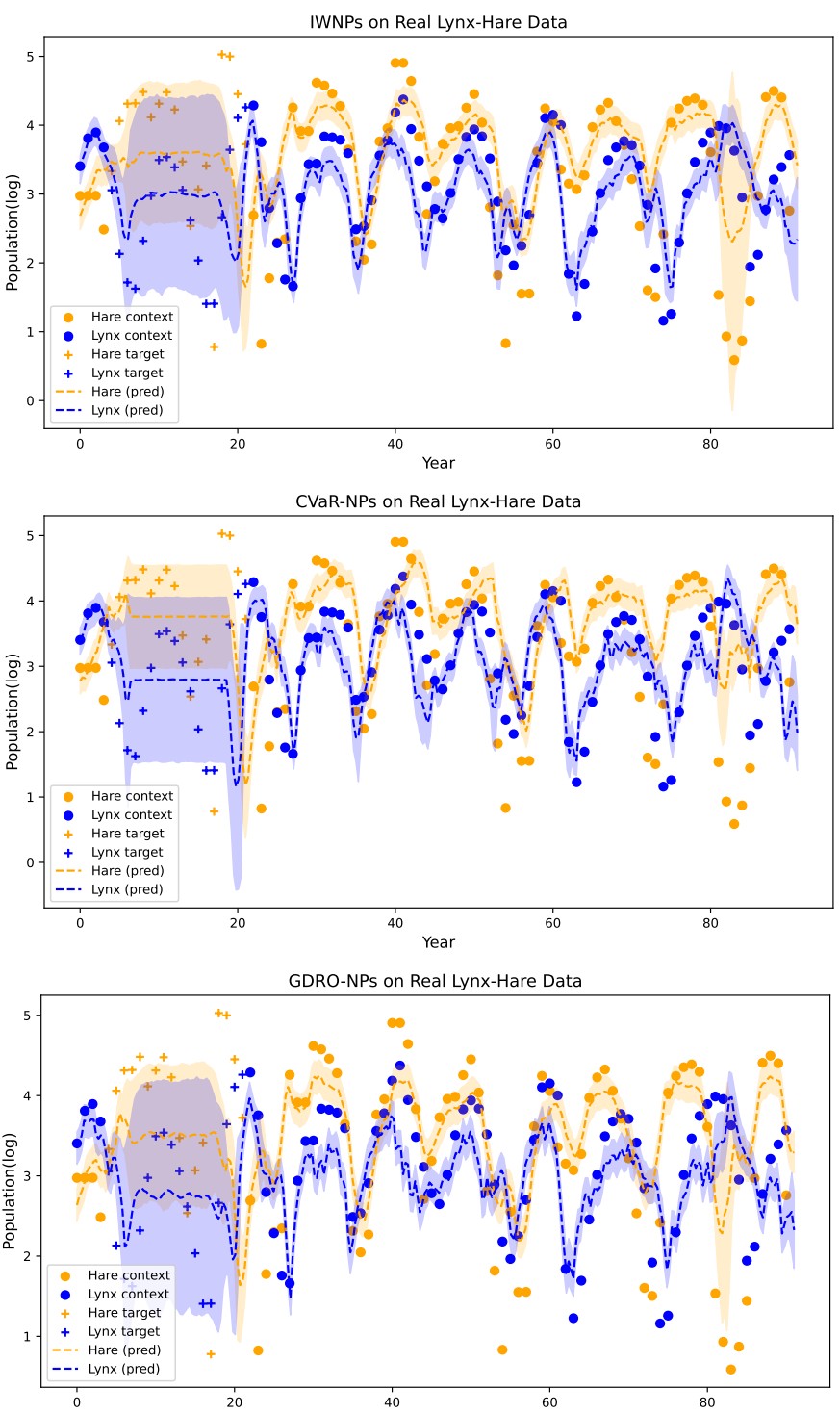

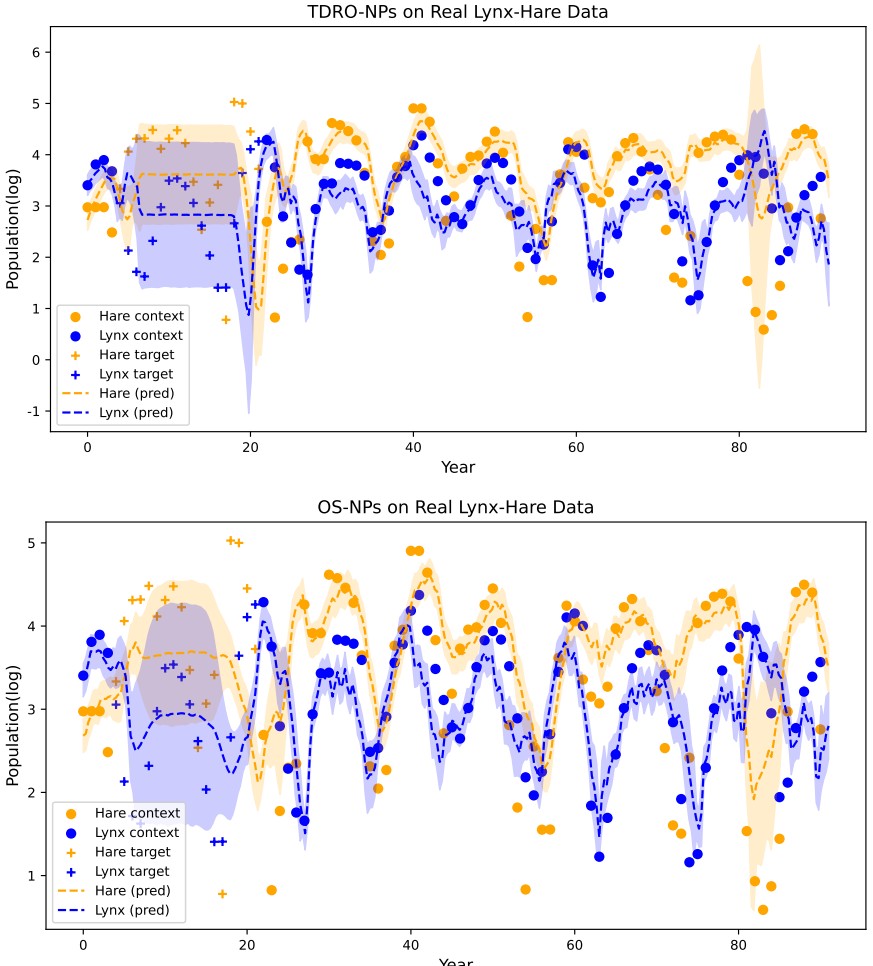

*Figure 11.* **Qualitative Sim-to-Real results for Lynx–Hare population interpolation.** The plots illustrate the predictive distributions of IWNPs, CVaR-NPs, GDRO-NPs, TDRO-NPs, and OS-NPs. Shaded regions represent $\pm 2$ standard deviations around the predictive mean, highlighting the models' uncertainty quantification and interpolation capabilities under sim-to-real shift.

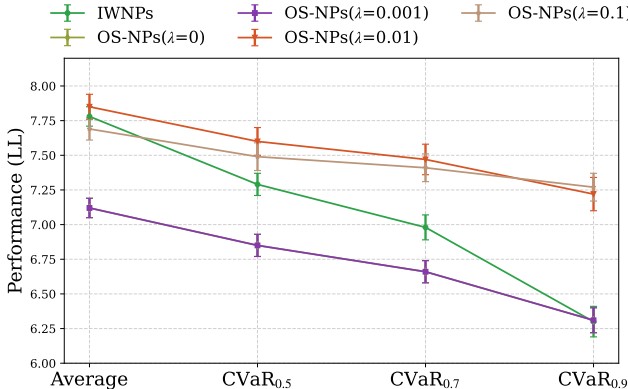

*Figure 12.* Choice of regularization coefficient $\lambda$ for OS-NPs vs. IWNPs baseline.

*Table 9.* Predictive log-likelihood on Fashion-MNIST, CIFAR-10, and SVHN (MLP architectures).

| Method | FMNIST | | | | CIFAR-10 | | | | SVHN | | | |
|---|---|---|---|---|---|---|---|---|---|---|---|---|
| | Avg | $\text{CVaR}_{0.5}$ | $\text{CVaR}_{0.7}$ | $\text{CVaR}_{0.9}$ | Avg | $\text{CVaR}_{0.5}$ | $\text{CVaR}_{0.7}$ | $\text{CVaR}_{0.9}$ | Avg | $\text{CVaR}_{0.5}$ | $\text{CVaR}_{0.7}$ | $\text{CVaR}_{0.9}$ |
| IWNPs | $\underline{0.94}_{\pm 0.001}$ | $0.76_{\pm 0.003}$ | $0.66_{\pm 0.004}$ | $0.48_{\pm 0.004}$ | $2.31_{\pm 0.006}$ | $2.17_{\pm 0.010}$ | $2.10_{\pm 0.011}$ | $1.96_{\pm 0.013}$ | $\mathbf{3.50}_{\pm 0.002}$ | $\mathbf{3.39}_{\pm 0.003}$ | $3.34_{\pm 0.004}$ | $3.23_{\pm 0.005}$ |
| CVaR-NPs | $0.93_{\pm 0.002}$ | $\mathbf{0.92}_{\pm 0.002}$ | $\mathbf{0.92}_{\pm 0.002}$ | $\mathbf{0.92}_{\pm 0.002}$ | $2.42_{\pm 0.009}$ | $\mathbf{2.42}_{\pm 0.009}$ | $\mathbf{2.42}_{\pm 0.009}$ | $\mathbf{2.42}_{\pm 0.009}$ | $3.40_{\pm 0.003}$ | $\mathbf{3.39}_{\pm 0.003}$ | $\mathbf{3.39}_{\pm 0.003}$ | $\mathbf{3.39}_{\pm 0.003}$ |
| GDRO-NPs | $0.89_{\pm 0.001}$ | $\underline{0.89}_{\pm 0.002}$ | $0.89_{\pm 0.002}$ | $\underline{0.89}_{\pm 0.002}$ | $2.33_{\pm 0.008}$ | $2.33_{\pm 0.008}$ | $\underline{2.32}_{\pm 0.008}$ | $\underline{2.32}_{\pm 0.008}$ | $3.34_{\pm 0.003}$ | $3.34_{\pm 0.003}$ | $3.33_{\pm 0.003}$ | $\underline{3.33}_{\pm 0.003}$ |
| TDRO-NPs | $\mathbf{0.96}_{\pm 0.001}$ | $0.83_{\pm 0.002}$ | $0.76_{\pm 0.003}$ | $0.63_{\pm 0.004}$ | $\underline{2.44}_{\pm 0.007}$ | $2.32_{\pm 0.008}$ | $2.26_{\pm 0.009}$ | $2.14_{\pm 0.010}$ | $3.44_{\pm 0.002}$ | $3.32_{\pm 0.003}$ | $3.27_{\pm 0.004}$ | $3.18_{\pm 0.005}$ |
| OS-NPs(Ours) | $\mathbf{0.96}_{\pm 0.002}$ | $\mathbf{0.92}_{\pm 0.002}$ | $\underline{0.90}_{\pm 0.003}$ | $0.86_{\pm 0.005}$ | $\mathbf{2.45}_{\pm 0.007}$ | $\underline{2.37}_{\pm 0.009}$ | $\underline{2.32}_{\pm 0.010}$ | $2.22_{\pm 0.011}$ | $\underline{3.46}_{\pm 0.002}$ | $\underline{3.38}_{\pm 0.003}$ | $\underline{3.35}_{\pm 0.003}$ | $3.27_{\pm 0.004}$ |

## E.1. Configuration of All Methods

For a fair comparison, we evaluate all baselines and our proposed method under a unified experimental framework. The general setup is same as (Foong et al., 2020) for all methods.

The vanilla IWNP follows the standard setup in (Foong et al., 2020) and implements the pipeline adapted from `https://github.com/YannDubs/Neural-Process-Family`. The GDRO implementation follows the reweighting setup described in `https://github.com/kohpangwei/group_DRO`, and we directly apply it to IWNPs. Specifically, we employ a group weight adjustment rate of 2 across all experimental configurations. For CVaR-NPs, it is a subset selection method (Zhang et al., 2022; Sun et al., 2026; Wang et al., 2026) and this work adopts a Monte Carlo estimate of $\text{CVaR}_{0.5}$ as the default for all methods, which means we sample 32 particles and choose the 16 worst particle-generated results to optimize. In terms of TDRO-NPs in Eq. (55), the temperature parameter is set to be $\lambda = 10$ for all benchmarks. For all experiments, including synthetic regression, image completion, and sim-to-real setups, we set the $\ell_2$ regularization coefficient $\lambda = 0.01$. This configuration was maintained throughout our ablation studies evaluating the sensitivity of OS-NPs to varying numbers of strata $K$.

We have attached the demo Python code in the supplementary material. The full code will be released and made public after the final version is confirmed.

## E.2. Meta Learning Datasets

**Synthetic Regression.** Following the experimental protocols established in (Foong et al., 2020) and (Gordon et al., 2019), we synthesize 1D stochastic functions via Gaussian process (GP) simulation. For each individual task, input coordinates $x$ are sampled from $\mathcal{U}[-2, 2]$ with a constant zero-mean prior. The context set size is determined by $n \sim \mathcal{U}[3, 47]$, while target points are allocated such that the total set size $n + m$ reaches up to 50. To ensure a diverse functional space, we utilize three distinct covariance structures with randomized hyperparameters:

- Matern $-\frac{5}{2}$ kernel: Captures functions with intermediate smoothness using

$$k(x, x') = \sigma^2 \left(1 + \frac{\sqrt{5}d}{\ell} + \frac{5d^2}{3\ell^2}\right) \exp\left(\frac{-\sqrt{5}d}{\ell}\right)$$

  where $d = 4|x - x'|$, $\sigma \sim \mathcal{U}[0.1, 1.0]$ and $\ell \sim \mathcal{U}[0.1, 0.6]$;

- Radial Basis Function (RBF) Kernel: Models infinitely differentiable, smooth variations defined by

$$k(x, x') = \sigma^2 \exp\left(-\frac{(x - x')^2}{2\ell^2}\right)$$

  with scale $\sigma \sim \mathcal{U}[0.1, 1.0]$ and lengthscale $\ell \sim \mathcal{U}[0.1, 0.6]$;

- Periodic kernel: Represents recurring patterns via

$$k(x, x') = \sigma^2 \exp\left(\frac{-2\sin^2\left(\frac{\pi\|x-x'\|^2}{p}\right)}{\ell^2}\right)$$

where the period $p \sim \mathcal{U}[0.1, 0.5]$ and remaining parameters follow the aforementioned uniform priors.

**Image Datasets.** Our empirical evaluation utilizes four standard image benchmarks: MNIST, Fashion-MNIST (FMNIST), CIFAR-10, and SVHN. During both the meta-optimization and evaluation phases, we define tasks by stochastically partitioning each image batch into context and target sets. Specifically, the context set cardinality $n$ is sampled from a discrete uniform distribution, where $n \sim \mathcal{U}[0, 784]$ for MNIST and FMNIST, and $n \sim \mathcal{U}[0, 1024]$ for CIFAR-10 and SVHN. Raw pixel intensities are mapped to a normalized tensor space using the PyTorch library. The remaining architectural configurations and training protocols strictly follow the experimental settings established in prior Neural Process literature (Garnelo et al., 2018a;b; Kim et al., 2019).

**Sim2Real Dataset.** To expand our evaluation, we incorporate the Sim2Real Benchmark, following the experimental pipeline and meta-learning configurations detailed by Gordon et al. (2019). The training regime utilizes synthetic trajectories generated from Lotka-Volterra equations, while performance is validated against empirical biological observations from the Predator-Prey dataset. We apply a global normalization step to all time-series data prior to the optimization phase to ensure numerical stability. Comprehensive specifications regarding the simulation parameters and dataset characteristics are available in the aforementioned reference.

### E.3. Neural Architectures & Optimizations & Evaluation Set-up

Across all experiments, we set $\lambda = \beta = 0.01$ and initialize $\xi^0 = \mathcal{L}^0 - \text{mean}(\mathcal{L}^0)$.

**Synthetic Regression.** For the Synthetic Regression experiments, we adopt the architectural configurations established by Gordon et al. (2019) and Foong et al. (2020) across all evaluated models to ensure a fair comparison. The model captures functional uncertainty through a 128-dimensional latent space. The Encoder (`conv_encoder`) is implemented as a deep set architecture with two hidden layers, each containing 128 units, while the Decoder (`conv_decoder`) utilizes a convolutional neural network with 128-unit hidden layers to induce spatial dependencies. Optimization is performed using the Adam optimizer with a fixed learning rate of $3 \times 10^{-4}$ over 200 meta-training iterations. Each training batch consists of 16 stochastically sampled tasks. To approximate the log-likelihood during meta-training, we employ 16 Monte Carlo particles for both importance-weighted and stratified variants (IWNPs and OS-NPs), increasing the sample count to 32 during the evaluation phase to ensure more precise predictive estimates.

**Image Completion.** Our implementation for image-based benchmarks adopts the on-the-grid ConvNP framework (Foong et al., 2020). The Encoder (`conv_encoder`) processes the concatenated context set and mask through an initial depthwise separable convolutional layer with a kernel size of 11, where positivity is enforced by convolving with the absolute value of the weights. This is followed by a ResNet backbone comprising 9 pre-activation residual blocks, each containing two depthwise separable convolutions and batch normalization. The resulting representation is mapped via a 128-dimensional latent space ($z \in \mathbb{R}^{128}$), which captures both local and global dependencies. Specifically, the Decoder (`conv_decoder`) employs a hybrid functional representation where half of the latent channels (64) are spatially mean-pooled to provide a global context, while the remaining channels preserve local pixel-dependent features. This combined representation is passed through a second 9-block ResNet and a final linear output layer to parameterize the predictive distribution. Optimization is performed with a fixed learning rate of $5 \times 10^{-4}$ and a batch size of 16 for all image benchmarks. We train models for up to 100 epochs on all image benchmarks, employing early stopping with a patience of 10 epochs. During the meta-training phase, the objective is approximated using 16 Monte Carlo particles for IWNPs and our stratified approach (OS-NPs). For our proposed method, we configure 2 boxes for MNIST and Fashion-MNIST, and 4 boxes for CIFAR-10 and SVHN. Additionally, a restart mechanism is employed to re-initialize the $\xi$ update at different intervals: every 5 epochs for CIFAR-10 and MNIST, and every epoch for SVHN and Fashion-MNIST. For evaluation, the sampling density is increased to 32 particles across all latent variable models.

**Sim2Real.** To ensure experimental consistency, our implementation for Sim2Real tasks follows the architectural and procedural protocols established by Gordon et al. (2019) and Foong et al. (2020). The network configurations mirror those utilized in our synthetic regression experiments, modified only to expand the output space to a 2-dimensional vector. Additionally, we employ a restart mechanism that re-initializes the $\xi$ update every epoch. For a comprehensive specification of auxiliary hyperparameters, we refer readers to Gordon et al. (2019). For evaluation, the sampling density is increased to 32 particles across all latent variable models. Performance is assessed by computing average log-likelihoods over 18 randomly selected consecutive data points from the Real Lynx-Hare dataset (1845–1935).

**Evaluation Set-up of Datasets.** In accordance with the benchmarking procedures established by Gordon et al. (2019),

Foong et al. (2020), Nguyen & Grover (2022), and Venkataramanan & Denzler (2025), our evaluation phase utilizes a stochastic task-partitioning strategy. For each batch during inference, we dynamically sample context set cardinalities within the intervals specified in the preceding sections. The reported performance metrics represent the aggregate average across all stochastically generated test batches. This evaluation protocol ensures that our results accurately reflect model generalization across varying levels of information density, consistent with established literature.

