# OpenReview forum: "Latent Space Robust Optimization of Neural Processes with Aligned Stratified Order-Statistic Loss Reduction"
_ICML.cc/2026/Conference — ICML 2026 regular_

### Official Review · Reviewer_p7gJ · 2026-03-06

**Soundness:** 3
**Presentation:** 3
**Significance:** 2
**Originality:** 3
**Overall Recommendation:** 4
**Confidence:** 4

**Summary:**

The paper puts forward OS-NPs, a method for optimizing neural processes inspired by multi-objective optimization (MOO). The authors highlight two undesired behaviors in the optimization of neural processes with multiple particles: IWNPs, which place too much emphasis on the most dominant particles and may therefore neglect others, and CVaR-NPs, which emphasize the worst-performing particles at the expense of average performance. To bridge these requirements, the authors adopt a multi-objective optimization perspective in which each region of the loss distribution represents a different task. They propose a lightweight variant based on MGDA, a known method for MOO, that aims to balance improvements across multiple regions. The proposed objective maintains desirable properties, such as the uniqueness and existence of optimal loss weights per iteration and convergence of the optimization process. Results show clear improvements for OS-NPs over baseline methods.

**Compliance With Llm Reviewing Policy:**

Affirmed.

**Final Justification:**

The rebuttal addressed my main concerns, mainly the additional experiments, including best particle performance and comparison on ImageNet.

**Key Questions For Authors:**

.

**Limitations:**

Yes, but it was in the Appendix and quite vague. I believe the proper place for it is in the main text.

**Strengths And Weaknesses:**

Strengths:
* The key strength of this paper, in my opinion, is the connection between the importance-weight objective and multi-objective optimization. This idea is novel and creative. It also opens an avenue for future research using more recent MOO techniques that may be more performant than MGDA.
* Both the empirical claims about IWNP behavior and the theoretical results in deriving the final objective in Eq. 18 and providing convergence guarantees, seem solid and well suited to conveying the paper’s main message.
* The empirical results on the evaluated benchmarks appear strong and clearly demonstrate the method’s advantage.

Weaknesses:
* The main weakness of this paper, in my opinion, is the empirical evaluation. First, the benchmarks used are quite outdated. I believe more complex image datasets (e.g., ImageNet) should be included to better showcase the method’s advantages and to advance the field more broadly. Second, I would like to see more examples of image completion, not only for worse-performing particles, but also for the best-performing ones.
* Significance: currently the comparisons in the paper are to other neural process approaches. However, to better position this paper it would be interesting to compare and clarify the main advantages of using OS-NP over other research attempts. For instance, the image completion task can be done with diffusion models as solving the inverse problem of inpainting.
* Some parts of the writing could be improved. For example, without looking at Figure 3, it is unclear whether the binning is applied at each step.
* Typos / corrections:
   * The definition of $d^t$ in Eq. 12 is missing the gradient.
   * Line 216 refers to Eq. 10b, which does not exist.

---

> ### Author Rebuttal · Authors · 2026-03-30
>
> We thank **Reviewer #p7gJ** for recognizing *(i) the novelty and creativity of establishing a connection between the importance-weight objective and MOO, (ii) solid theoretical results with convergence guarantee, and (iii) strong and clear empirical results*. We are also grateful for your constructive and insightful suggestions. Below, we address the specific concerns.
>
> ---
> ### **1. Scalability in more complex dataset (W1)**
>
> Thanks for the comment.
> **(1) Standard Benchmarks:** We initially adopted standard and commonly-used robust meta-learning and NP benchmarks [1-3], utilizing few-shot curve fitting, image completion, and Sim2Real to demonstrate the versatility of OS-NPs.
> **(2) New mini-ImageNet Experiments:** Following your excellent suggestion to include higher-complexity benchmarks, we evaluated our method on mini-ImageNet. Maintaining the exact setup and protocols used for CIFAR-10/SVHN, we report the meta-testing completion log-likelihoods and training time per epoch below:
>
> | Methods   | Average   | CVaR$_{0.5}$ | CVaR$_{0.7}$ | CVaR$_{0.9}$ |
> |-----------|-----------|------------|------------|------------|
> | IWNPs     | 5.519     | 4.135      | 3.449      | 2.275      |
> | CVaR-NPs  | 6.404     | 6.406      | 6.402      | **6.4**    |
> | GDRO-NPs  | 2.12      | 2.119      | 2.119      | 2.118      |
> | TDRO-NPs  | 3.912     | 3.602      | 3.416      | 3.0        |
> | OS-NPs    | **6.895** | **6.678**  | **6.559**  | 6.278      |
>
>
> | Methods                   | IWNPs | CVaR-NPs | GDRO-NPs | TDRO-NPs | OS-NPs |
> |---------------------------|-------|----------|----------|----------|--------|
> | Training time(mins/epoch) | 19.51 | **33.42**   | 20.33    | 19.50    | 21.08  |
>
>
> These results confirm that OS-NPs maintain their average and tail-risk advantages on more complex data. While CVaR-NPs achieve superior robustness due to mini-ImageNet's out-of-distribution (OOD) setting (non-overlapping classes), they require 1.5x more training time than OS-NPs.
>
> **Reference:**
>
> [1] Translation Equivariant Transformer Neural Processes. ICML2024.
> [2] Distance-informed Neural Processes. NeurIPS2025.
> [3] Learning Robust Neural Processes with Risk-Averse Stochastic Optimization, ICML25
>
> *We’ll add the above result and discussion to **Section 4.2** in the updated paper.*
>
> ### **2. Analysis on Examples of Best-Performing Particles (W1)**
>
> **(1) Primary Focus and Scope:** This work's topic is inducing latent space robustness to improve the generation quality of worst-case particles, which naturally aligns with our regularized MOO formulation.
>
> **(2) Best-Case Evaluation:** Valuing best-performing completions, we evaluate the top 10% of particles for mini-ImageNet visualizations and CIFAR-10 predictive LL. Comparative results are available at https://anonymous.4open.science/r/system-identification-153C, where OS-NPs outperform all baselines when evaluated on the best-performing particles.
>
> *We’ll include the above analysis of best-performing particles in **Section4.2**.*
>
>
> ### **3. Comparison with Diffusion-based Models (W2)**
>
> Thank you for this insightful comment. We agree that diffusion models excel at image completion. However, OSNP fundamentally differs in scope and motivation:
>
> **(1) Versatile Few-Shot Focus:** NPs are versatile probabilistic learners designed for diverse domains (e.g., regression, control, vision), not for chasing application-specific SoTA in image generation.
>
> **(2) Latent Architecture:** Diffusion models use hierarchical latents, which complicates defining latent robustness in this manner. OS-NPs operate on a single, explicit latent space.
>
> **(3) Targeted Problem:** The Matthew effect is a pathology specifically inherent to IWNPs. As noted by **Reviewers #niCy**, core contribution lies in providing a "targeted solution" to this exact structural drawback within NP.
>
>
> ### **4. Presentation Suggestion, Typos (W3)**
>
> Thanks for pointing out the typos and writing suggestions. We apologize for these mistakes and address the issues below:
>
> (1) **Binning description**: We’ll add a description sentence around Algorithm 1:
>
> >At each optimization step, we sample B particles from the prior, compute their per-particle losses, sort them, and partition them into K disjoint strata based on their order statistics.
>
> (2) **Eq.12 polish**: We’ll add the missing gradient term as:
> $\theta^{t+1}=\theta^t-\eta d^t,\
>     \text{with}\
>     d^t=\sum_{i=1}^{K}\alpha_{i} \nabla_{\theta} \mathcal{L}_{i}(\theta).$
>
> (3) **Line 216**: The reference to "Eq. 10b" has been corrected to "Eq.10a”.
>
> (4) We’ll polish the limitation as:
>
> >Our amortized approximation in Eq. (17) induces an approximation error. The error can be bounded over iterations, as proved in Proposition 3.2. However, it can still be a barrier that hinders the model from achieving its best performance. (Move to main paper)
>
> ---
> *Thanks for your efforts. We hope concerns are well addressed. It would be appreciated if you can reconsider the evaluation.*

---

> > ### Author Rebuttal · Reviewer_p7gJ · 2026-04-02
> >
> > I thank the authors for their response, which addressed my concerns. In light of it, I will raise my score by one point.

---

> > > ### Author Response · Authors · 2026-04-02
> > >
> > > We appreciate Reviewer p7gJ and Area Chair's efforts during the rebuttal. These suggestions and questions are precious for our manuscript improvement. We'll incorporate them into the update version. Thanks.

---

### Official Review · Reviewer_niCy · 2026-03-09

**Soundness:** 4
**Presentation:** 4
**Significance:** 3
**Originality:** 3
**Overall Recommendation:** 5
**Confidence:** 4

**Summary:**

The authors demonstrate that standard optimization of Importance-Weighted Neural Processes suffers from the Matthew effect. Existing solutions can mitigate such risks of poor tail generalization and unstable adaptation, but at the cost of average performance. This paper proposes Order-Statistics Aligned Neural Processes (OS-NPs) to mitigate the risks entailing the Matthew effect while not compromising on the average performance. This behavior is achieved by stratification of the losses of each particle, and applying the optimization within each stratum jointly. The most important contribution of the paper lies in developing a practical realization of the optimization problem as a result of this stratification, which is supported by sound theory and reasonable set of experiments.

**Compliance With Llm Reviewing Policy:**

Affirmed.

**Key Questions For Authors:**

* Line 92(right): Why is the ELBO broken? It might be worth expanding with a sentence or two for completeness of presentation.

**Limitations:**

Yes

**Strengths And Weaknesses:**

The paper is well-written and provides a clear series of sequential building blocks of theory. Each result is supported by reasonable discussion which gives me confidence that this paper will be a good read for the community.

The contributions can essentially be boiled down to solving the Matthew effect by instead converting the original optimization problem into a multi-objective optimization problem via stratification of the particle losses, and realizing a practical method for the same. All the expected ablations for the method are provided. I strongly suggest the authors to highlight these specifics earlier in the paper, instead of a generic 2-point contribution list since these details are significant.

The paper focuses on a single drawback of IWNP, and provides a targeted solution. There is a clearly trackable lineage of ideas that led to this paper, and I appreciate the authors for maintaining that in writing while distinguishing their own contributions. The authors provide an important insight into the behavior of neural processes, which is supported in an instructive manner via theory and a simple experiment. In this sense, this paper bears educational value as well.

---

> ### Author Rebuttal · Authors · 2026-03-30
>
> We sincerely thank **Reviewer #niCy** for the positive evaluation of our work, including *(i) important insight into NPs' behavior, (ii) sound theory, and (iii) a reasonable set of experiments*. Your suggestions are extremely helpful for improving the quality and impact of our manuscript.
>
> ---
>
> ### **1. Highlighting technical details earlier (Suggestion)**
>
> Thank you for this suggestion. A more descriptive presentation of our contributions will surely better serve readers. In the revised manuscript, *we'll polish the contributions summary in the **Introduction Section Line77-94** to explicitly highlight the core idea*. Specifically, updated contents are as follows:
>
> >Our primary technical contribution is two-fold:
>
> >*1. We formally identify a critical limitation of the importance-weighted optimization objective in NPs, where the gradient collapses to the best-performing particle, which we term the Matthew Effect. Accordingly, this work introduces the concept of fast adaptation robustness in the latent space for the NP family.*
>
> >*2. To balance the worst-case adaptation and average performance, we reframe the latent space optimization as a stratified multi-objective optimization (MOO) problem. By deriving a regularized worst-case optimization strategy and solving it via an amortized gradient alignment technique, we craft Order-Statistics Aligned Neural Processes (OS-NPs) in a computationally efficient manner.*
>
>
> ### **2. More descriptions on the broken ELBO (Key Questions in Line 92).**
>
> We appreciate this suggestion for improving completeness. *We'll include a brief description sentence in **Maximum-Likelihood in IWNPs (Line 91)** to explain this point*. The new contents will be added in **Line 94**:
>
> >*Note that there is no exact form of functional prior or posterior. Garnelo, et al. [1] introduce a surrogate objective, which is often referred to as a 'broken' ELBO. One shared neural network in vanilla NPs parameterizes both the approximate posterior and the prior, leading to a training objective that does not strictly adhere to the variational inference framework.*
>
> **References:**
>
> [1] Neural processes. ICML 2018 workshop.
>
> ---
>
> *Finally, we express gratitude to **Reviewer #niCy**'s constructive and insightful reviews once again. For other questions or suggestions, we are happy to discuss.*

---

### Official Review · Reviewer_kcr3 · 2026-03-26

**Soundness:** 3
**Presentation:** 3
**Significance:** 3
**Originality:** 2
**Overall Recommendation:** 4
**Confidence:** 2

**Summary:**

The paper identifies a Matthew effect in Importance-Weighted Neural Processes (IWNPs), where high-likelihood inference particles disproportionately dominate the gradient signal. To mitigate this, the authors propose Order-Statistics Aligned Neural Processes (OS-NPs). This approach sorts inference particles into disjoint difficulty strata and applies an amortized, regularized multi-objective optimization framework to balance average-case and tail-risk performance.

**Compliance With Llm Reviewing Policy:**

Affirmed.

**Final Justification:**

The authors have addressed all of my concerns. So I raised my score to 4.

**Key Questions For Authors:**

- The paper effectively motivates the necessity of CVaR optimization by highlighting safety-critical domains, such as robotic systems, where low-quality, tail-risk generations can mislead policy optimization and result in dangerous actions. However, the current evaluations primarily rely on standard vision benchmarks (FMNIST, CIFAR-10, SVHN) and synthetic Gaussian Process tasks. Do you have specific safety-focused evaluations, reinforcement learning environments, or critical out-of-distribution stress tests where OS-NPs demonstrably prevent catastrophic downstream failures compared to standard IWNPs?

**Limitations:**

yes

**Strengths And Weaknesses:**

## Strengths

- The submission is theoretically well-grounded. The authors mathematically formalize the "Matthew effect" in Importance-Weighted Neural Processes (IWNPs), proving how gradient signals collapse to the single best-performing particle. Furthermore, the proposed method is supported by a rigorous convergence analysis for the stratified gradient descent and a proof of strict strong convexity for their regularized inner-loop objective.

- The paper tackles a relevant problem in probabilistic meta-learning: improving generation fidelity and tail-risk robustness in risk-sensitive applications (such as robotics) where poor tail-risk generation can mislead policy optimization.

- The paper is clearly written, and the narrative flows logically from diagnosing the IWNP optimization imbalance to detailing the multi-objective solution. The authors are also commendably transparent about the computational overhead, providing a theoretical complexity analysis and wall-clock time comparisons in the appendix.

## Weaknesses

- The empirical evaluation undermines the practical soundness of the method. While the bi-level optimization pipeline is technically correct, it introduces substantial algorithmic complexity via particle sorting, histogram binning, and multi-objective gradient aggregation. However, the performance gains are incredibly marginal. For instance, in Table 2, the FMNIST average log-likelihood only increases from 2.81 (IWNPs) to 2.85 (OS-NPs). Similarly, the SVHN average metric only improves from 9.27 to 9.29. While tail-risk (CVaR) metrics show slightly better relative improvement, it is difficult to justify the heavy optimization machinery for such minor baseline shifts.


- The presentation downplays the practical friction of implementing and tuning this method. While the authors note that the actual wall-clock time overhead is relatively small (e.g., 6.27 minutes per epoch for OS-NPs versus 5.43 minutes for IWNPs on CIFAR-10) , the method requires practitioners to manage a nested loop and tune multiple new hyperparameters, including the number of strata $K$, the regularization coefficient $\lambda$, and the inner-loop learning rate $\beta$. A more honest discussion regarding the cost-benefit trade-off of this tuning burden would improve the paper.

- While the application to the latent space of NPs is novel, the underlying algorithmic components, such as histogram binning , CVaR optimization , and amortized gradient manipulation, are heavily drawn from existing literature. The originality lies primarily in the combination of these established tools rather than a fundamental methodological breakthrough.

---

> ### Author Rebuttal · Authors · 2026-03-31
>
> We thank the reviewer for recognizing *(i) OS-NPs' theoretical groundedness and novelty, (ii) rigorous convergence analysis, and (iii) the relevance of tail-risk robustness*. Below, we address your specific concerns:
>
> ---
>
> ### **W1. Explanations on Performance Gains and Algorithmic Complexity**
>
> **(1) Robustness as the Primary Evaluation rather than Average Gains.**
> While average gains on standard benchmarks appear marginal, **our primary focus is latent space robust adaptation**. We resolve the IWNP Matthew effect to enhance worst-particle performance without sacrificing average metrics, a common flaw in CVaR Meta-Learning [1-3]. From **Table 1-3**, OS-NPs achieve massive $\text{CVaR}_{0.9}$ improvements over IWNPs, e.g., **47%** (RBF), **59.3%** (SVHN), and **23%** (Sim2Real).
>
> **(2) Marginal Training Overhead for Significant Test-Time Robustness.**
> Added complexity (**Line 280**) is $\mathcal{O}(B \ln B + K)$, minimal vs. forward/backward passes. Exchanging marginal training compute for significant inference robustness is highly worthwhile. (*Will clarify this in Sec. 4*).
>
> ### **W2. Practical Implementation and Tuning Friction Discussion**
>
> We’ll take your advice to clarify implementation frictions and add the discussions below.
>
> **(1) Lightweight Plug-and-Play Integration & Favorable Cost-Benefit Trade-off.**
>
> > (See Code) Upgrading to OS-NPs introduces minimal friction, requiring only simple binning and amortized modules. Marginal overhead is strictly confined to meta-training. At test-time, OS-NPs retain standard IWNPs' fast-adaptation efficiency. （Will add to **Sec. 3.3**）
>
> **(2) No Extensive Hyperparameter Tuning Required.**
>
> > OS-NPs avoid heavy tuning burdens. We used $K=4$ strata across all benchmarks, a stable "plug-and-play" setting validated by our ablation study (Fig. 7). Similarly, we fixed regularization ($\lambda$) and inner-loop learning rate ($\beta$) at $0.01$ globally (App. E.3). We intentionally restricted exhaustive tuning to demonstrate fair, out-of-the-box utility. (Will add to **Sec. 4**)
>
> ### **W3. Primary Contribution Clarification**
>
> **(1) Theoretical Novelty:** While binning/CVaR exist, our core contribution (**Lines 77-90**) is the theoretical framework: the regularized MOO in Eq. (16) solving the IWNP bottleneck, supported by Prop 2.1 & Thm 3.1-3.2. **Reviewers #niCy** and **#p7gJ** confirm these novelties.
>
> **(2) Principled Derivation:** By mathematically formalizing the IWNP "Matthew Effect," our algorithmic pipeline naturally emerges as the required mathematical solution, rather than a mere empirical combination.
>
> ### **Q1. RL Environment as New Benchmarks and OOD Stress Test**
>
> **(1) New RL Env Evaluations:** Following your advice, we conducted safety-critical tests on meta-learning RL env transition dynamics with the same setup of Pendulum System in [3]. During meta-training, physical parameters (mass $m$, length $\ell$) are sampled from $[0.4, 1.6]^2$. To assess OOD generalization, we tested on unseen, extrapolated dynamics shifted to $[1.7, 2.5]^2$. Results see https://anonymous.4open.science/r/system-identification-153C.
>
> | Pendulum-ID | Average      | CVaR$_{0.5}$ | CVaR$_{0.7}$ | CVaR$_{0.9}$ |
> |----------|--------------|--------------|--------------|--------------|
> | IWNPs    | 0.591 ± 0.02 | 0.659 ± 0.03 | 0.733 ± 0.06 | 0.780 ± 0.06 |
> | OS-NPs   | 0.585 ± 0.02 | 0.652 ± 0.01 | 0.709 ± 0.02 | 0.727 ± 0.04 |
> | **Pendulum-OOD** | Average      | CVaR$_{0.5}$ | CVaR$_{0.7}$ | CVaR$_{0.9}$ |
> | IWNPs    | 0.621 ± 0.01 | 0.675 ± 0.03 | 0.751 ± 0.06 | 0.793 ± 0.04 |
> | OS-NPs   | 0.634 ± 0.03 | 0.669 ± 0.02 | 0.722 ± 0.04 | 0.756 ± 0.06 |
>
> As shown, OS-NPs mostly outperform IWNPs on in-distribution and OOD tasks across average and CVaR MSEs, preventing catastrophic dynamics prediction failures. We’ll add these evaluations to the revision. (Note: OOD signal completion in mini-ImageNet results are reported to **Reviewer #p7gJ**).
>
> **(2) Safety-Critical Nature of Existing Benchmarks:** We initially adopted standard and commonly-used robust meta-learning and NP benchmarks [1-4] because they inherently represent genuine safety-critical challenges. Worst-case curve fitting imitates trajectory prediction; partial image completion simulates sensor occlusions requiring robust generation for safe decisions; and our Sim2Real experiments (**Line 371, Table 3**) explicitly demonstrate OS-NPs' superior OOD robustness against unpredictable real-world dynamics.
>
> ### **References:**
>
> [1] Task-Robust Model-Agnostic Meta-Learning, NeurIPS20
> [2] A Simple Yet Effective Strategy to Robustify the Meta Learning Paradigm, NeurIPS23
> [3] Robust fast adaptation from adversarially explicit task distribution generation, KDD25.
> [4] Learning Robust Neural Processes with Risk-Averse Stochastic Optimization, ICML25
>
>
> ---
> *Finally, thanks again for your efforts and insightful comments. We hope your concerns are well addressed. It would be appreciated if you can reconsider the evaluation of our work.*

---

> > ### Author Rebuttal · Reviewer_kcr3 · 2026-04-01
> >
> > All of my concerns are resolved.

---

> > > ### Author Response · Authors · 2026-04-01
> > >
> > > We thank **Reviewer #kcr3** and the **Area Chair**'s efforts and constructive suggestions. We will incorporate all revisions in the updated manuscript, and these suggestions significantly improve the quality of our work.

---

### Decision · Program_Chairs · 2026-04-30

**Decision:**

Accept (regular)

**Comment:**

This paper studies a pathology of importance-weighted neural process (IWNP) training, referred to as the Matthew effect, where a single high-likelihood particle dominates the gradient signal. To address this issue, the authors propose a principled approach that reformulates the IWNP objective as a stratified multi-objective optimization problem based on order statistics. Several reviewers find this perspective novel and insightful.

There were initial concerns regarding the empirical evaluation, including the magnitude of performance gains and the choice of benchmarks, as well as the added algorithmic complexity. These concerns were largely addressed during the rebuttal, with additional experiments and clarifications. Overall, I align with the reviewers’ assessment, particularly regarding the novelty and conceptual clarity of the proposed approach.